# A micro-dispenser for long-term storage and controlled release of liquids

Amin Kazemzadeh[1], Anders Eriksson [2], Marc Madou[3] & Aman Russom[1]

The success of lab-on-a-chip systems may depend on a low-cost device that incorporates on-chip storage and fluidic operations. To date many different methods have been developed that cope separately with on-chip storage and fluidic operations e.g., hydrophobic and capillary valves pneumatic pumping and blister storage packages. The blister packages seem difficult to miniaturize and none of the existing liquid handling techniques despite their variety are capable of proportional repeatable dispensing. We report here on an inexpensive robust and scalable micro-dispenser that incorporates long-term storage and aliquoting of reagents on different microfluidics platforms. It provides long-term shelf-life for different liquids enables precise dispensing on lab-on-a-disc platforms and less accurate but proportional dispensing when operated by finger pressure. Based on this technology we introduce a method for automation of blood plasma separation and multi-step bioassay procedures. This micro-dispenser intends to facilitate affordable portable diagnostic devices and accelerate the commercialization of lab-on-a-chip devices.

---

[1] Division of Nanobiotechnology, Department of Protein Sciences, Science for Life Laboratory, KTH Royal Institute of Technology, Stockholm 17165, Sweden. [2] School of Engineering Sciences, Mechanics, KTH Royal Institute of Technology, Stockholm SE-10044, Sweden. [3] Department of Mechanical and Aerospace Engineering, University of California, Irvine, California 92697, USA. Correspondence and requests for materials should be addressed to A.K. (email: amink@kth.se) or to A.R. (email: aman@kth.se)

Two problems have largely kept microfluidic devices in research institutions and significantly limited their acceptance into the open market, i.e., the lack of generic proportional valves/pumps that act like micro-dispensers (1) and the lack of efficient long-term liquid storage (2). A distinctive reliable solution to these problems would integrate common microfluidics tasks such as valving, storing, and metering all in one device. Techniques that are being developed to address these listed microfluidic tasks have continuously evolved but are generally tackling each fluidic function separately. A comprehensive overview of the advances in developing fluid handling techniques and methods for operating various microfluidics tasks in lab-on-a-chip (LOC) devices can be found in several review articles[1–3]. Briefly, fluid handling in microfluidics can be performed either by employing passive or active techniques[1,4]. The passive methods only use forces related to the physical properties of the fluidic platform itself, e.g., capillary valves[5–7]. These techniques are usually susceptible to fabrication and material imperfections as well as to abrupt movement and vibrations, especially for highly wetting liquids. The active liquid handling techniques involve external power sources such as laser beams[8] or finger pressure[9,10]. Active valves actuated, say by a heat source or laser beam, increase the cost and add complexity to the microfluidics system[11,12]. Currently, a large variety of active and passive fluid handling techniques have been developed and used together in order to carry out different fluidic tasks. Aliquoting and metering are generally performed in few consecutive steps by employing downstream micro-structures and/or valves. For example, in lab-on-a-disc (LOD) platforms, reagents are first measured by being divided into defined sub-volumes before transferring to different destinations[1,13–15]. Using additional micro-chambers and valves enables sequentially and time-dependently release of liquids in theses platforms[16]. These and similar approaches used in LOC devices occupy huge space, complicate manufacturing, and increase fabrication costs. An alternative and expensive aliquoting method is to connect automatic dispensers to microfluidics devices using special pipette tips and supporting structures[15]. Similar to aliquoting, separation of molecules, cells, or solid particles from each other and from the surrounding medium is also often conducted by using distinct micro-structures according to the physical or chemical properties of the entities to be separated. In a LOC platform, the separation is often based on filtration, i.e., using a physical filter component or sedimentation[17,18]. Separating plasma from blood cells, especially in LOD platforms, is often performed using a siphoning mechanism, which requires local surface treatment and comparatively spacious metering structures[18–20].

The long-term storage of liquid reagents on a fluidic platform perhaps has been the most vexing of the listed problems[2]. Liquids can be stored either in locally structured enclosed chambers or in separately prepared cartridges that can be integrated with LOC devices. An example of pre-storage of liquids in LOC devices is using inert impermeable vessels such as glass or Teflon made and sealing them by ferrowax plugs[21]. For the release of reagents, the wax plug needs to be melted, which is usually done by a laser irradiation technique. We suggest that incorporating separately prepared containers into LOC devices may represent a more advantageous approach than fashioning valves and liquid storage devices on the platform itself. As such, glass ampules have been proposed in order to store liquids in LOD platforms[22]. They provide long shelf-life but require a separate release mechanism to break the ampules, which complicates automating the entire process. Another method of reagent storage on LOC devices is to fill and seal reagents in small packages, e.g., blister and stick packages[23,24]. A blister package usually consists of ultra-thin aluminum films protected by PET outer layers that enhance their handling and mechanical strength. This approach provides long-term reliable shelf-life but does not enable proportional release of reagents. The mean pressure needed to burst these packages is ~170 kPa and their miniaturization seems to be limited[25]. The content stored in these blisters is released more efficiently when a mechanical actuator is employed, which of course adds complexity, costs, and occupies space[24]. These packages connected to special receiving chamber that is equipped with a membrane valve enable multiple releasing of reagents[25,26]. However, it does not enable proportional release, adds costs, complexity, and makes a more specific approach. The blister seems difficult to miniaturize and permits reagent loss both due to residuals in the package and during transfer of reagent from the packages to downstream micro-structures[25].

These numerous attempts at separately developing various liquid handling and storage techniques are perhaps a response to the disappointing results in developing a single integrated device able to act as a valve, storage, and dispenser[4,11,27–30]. A system with robust valve, pump, and storage capabilities can accomplish automation of a large variety of clinical assays[31]. The development of a lab-in-a-tube (LIAT) system that consists of several mechanical parts and specifically made components is an example of such systems[31]. However, the solution to the above-mentioned problems must be robust, low-cost, integrated, and provide long-term liquid storage, proportional liquid dispensing as well as being compatible across different fluidic platforms. Without a satisfactory resolution of both proportional controlled release and on-chip storage of liquids, LOC devices will probably never meet the requirements for a successful point-of-care (POC) tests[2,32–37]. Here, we introduce a technology that eliminates most downstream micro-structures and micro-components by combining long-term liquid storage, valving/pumping, and proportional reagent dispensing. This technology intends to substantially reduce the fabrication costs and complexity of LOC devices and enables developing low-cost POC devices. It consists of a simple micro-dispenser that performs different microfluidics operations without sacrificing scalability and/or compatibility across different fluidic platforms. Our experiments verify our technology's efficiency, and numerical simulations support our simplified mechanical model that gives a general guide for designing the proposed micro-dispensers.

## Results

**Dispensing characteristics.** The accuracy of the proportional liquid dispensing for both linear and non-linear microfluidics and the long-term storage stability of the micro-dispenser were separately evaluated for more than 70 micro-dispensers. In order to test the dispensing accuracy of the micro-dispenser on a LOD platform, 21 different micro-dispenser devices were fabricated and placed into specifically designed compartments. Details of the experimental setup for constructing LOD platforms can be found in several published articles[3,28]. Here, we show the dispensing accuracy of a set of four micro-dispensers filled with 95% di-water, i.e., ~5% air volume. The micro-dispensers are made by piercing a hole on FEP sealed tubes of outer and inner diameter of 2 mm and 1 mm, respectively that contains 22 μl di-water. The apertures on the tubes are covered by latex membranes of outer and inner diameter of 2.4 mm and 0.8 mm, respectively. In order to measure the volume dispensed at each discharge, we measured the change in liquid volume inside the micro-dispensers after each discharge, see Supplementary Note 1 and Supplementary Figure 1. Note that the amount of liquid dispensed at a given pressure for different micro-dispenser configurations ranges from nano to micro liters, which depends on the amount of air inside the micro-dispenser, the elastic properties of the aperture

**Table 1 Micro-dispenser dispensing accuracy in lab-on-a-disc platforms**

| Sample no. | 285×g (nl) | 306×g (nl) | 329×g (nl) | 352×g (nl) | 376×g (nl) | 400×g (nl) | 426×g (nl) | Sr (nl) |
|---|---|---|---|---|---|---|---|---|
| 1 | 70–75 | 70–75 | 70–75 | 80–85 | 80–85 | 70–75 | 85–90 | 1 |
| 2 | 70–75 | 70–75 | 70–75 | 70–75 | 70–75 | 70–75 | 70–75 | 0.2 |
| 3 | 70–75 | 70–75 | 70–75 | 70–75 | 70–75 | 70–75 | 70–75 | 0.2 |
| 4 | 70–75 | 70–75 | 70–75 | 70–75 | 70–75 | 70–75 | 70–75 | 0.2 |
| s.d. | 4 | 4 | 4 | 7.1 | 7.1 | 2.9 | 9.5 | – |
| c.v._max (%) | 2.4 | 2.4 | 2.4 | 11.5 | 9.4 | 4 | 12.4 | – |
| s.e.m. (%) | 2.5 | 2.5 | 2.5 | 6.1 | 6.1 | 2.5 | 8.2 | – |

The experimental data for four sets of similar micro-dispensers, approximate dispensed volume corresponding to different rotational frequencies
s.d. standard deviation, c.v. coefficient of variation, s.e.m. standard deviation of mean, Sr repeatability standard deviation

**Table 2 Micro-dispenser dispensing accuracy in pressure-driven systems**

| Sample no. | Actuation pressure (kPa) | n = 1 (µl) | n = 2 (µl) | n = 3 (µl) | n = 4 (µl) | n = 5 (µl) | Mean (µl) | s.d. | s.e.m. | Sr (µl) | Ref. s.e.m. ± (µl)[38] | Ref. Sr ± (µl)[38] |
|---|---|---|---|---|---|---|---|---|---|---|---|---|
| 1 | 80 | 2.0 | 2.1 | 1.8 | 1.7 | 1.8 | 1.9 | 0.2 | 0.15 | 0.1 | 0.08 | 0.04 |
| 2 | 90 | 9.3 | 9.0 | 10.3 | 10.2 | 10.2 | 9.8 | 0.6 | 0.54 | 0.2 | 0.12 | 0.08 |
| 3 | 100 | 12.4 | 11.2 | 11.3 | 11.2 | 11.2 | 11.5 | 0.5 | 0.47 | 0.1 | 0.13 | 0.08–0.1 |

The experimental data for three sets of micro-dispensers
s.d. standard deviation, s.e.m. standard deviation of mean, Sr repeatability standard deviation

covering membrane, and the tightness of the sealing. The combination of membrane tightness and the low amount of air we used here allows for highly controllable and accurate liquid dispensing. In Table 1, we list consecutive and accurate aliquoting liquid from four micro-dispensers at different spinning frequencies together with the standard deviation errors and coefficient of variation. The results show that the micro-dispenser has comparable accuracy as conventional micro-dispensers in LOD systems[38]. We have obtained these data using image processing, see Supplementary Figure 1. Briefly, we have imported our experimentally obtained images to ImageJ and Photoshop software and investigated the liquid displacement in the micro-dispensers. In order to evaluate the aliquoting performance, we performed experiments for 3 sets of 3 micro-dispensers. The length, the inner and outer diameter of the micro-dispensers of each set remains the same, i.e., inner diameter of 1 mm and outer diameter of 1.9 mm. The length of the micro-dispenser is 29.5 ± 0.5 mm. The dispensing results presented in Table 2 are very similar at different spinning speeds due to the expansion of the air inside the micro-dispenser, which creates a negative pressure countering the centrifugal pressure that allows for consistent release of liquids when increasing spinning speed. Further, as we use image processing and manually measure the replacement of the liquid inside the micro-dispenser, there exists a certain amount of measurement error that could affect the results. As such, the data reported for aliquoting accuracy may benefit more when calibrated using sophisticated measurement methods/setups.

We investigated the dispensing accuracy on LOC systems by connecting the micro-dispensers to a pressure controller (Nagano Keiki, model no. PC20) and experimentally obtained the actuation pressure, i.e., the pressure at which the first droplet of liquid is dispensed. Suppose that the aperture has a centerline, in all sets the membrane asymmetrically covers the aperture, which allows unidirectional dispensing. The pressure increases gradually and the dispensed liquid is measured after ~30 s. Knowing this pressure, we investigated the dispensing accuracy of three sets of similar micro-dispensers under different pressures above the

actuation pressure. Table 2 lists the experimental data for the dispensing accuracy of three micro-dispensers, made from 1 ml syringes and a latex membrane with internal diameter and thickness of 3.2 mm and 0.8 mm, respectively. The discrepancy between liquid volumes dispensed in different micro-dispensers is rather due to the inequality in the length and the exact positioning of the membrane as they are cut and placed manually. In order to fully demonstrate the capability of our micro-dispenser in accurate and proportional dispensing of liquids, we compare our experimental results with a standard data sheet for conventional pipettes used in laboratories. The piston pipette standards published at the International Organization for Standardization (ISO 8655) defines that maximum acceptable measurement uncertainties for dispensing 1–10 µl, 10–100 µl, and 100–1000 µl are 0.05–0.12 µl, 0.12–0.8 µl, and 0.8–8 µl, respectively[38]. According to the same, acceptable coefficient of variation for the above-mentioned dispensing ranges are 5–0.8, 0.8–0.4, and 0.4–0.3. Knowing that these errors are twice as large for multichannel pipettes, values listed in Tables 1 and 2 show that the micro-dispenser has a great potential to operate within the range of acceptable measurement errors, especially for the use for LOD platforms when compared with the standard values.

**Actuation pressure**. Parameters of the micro-dispensers that can be used to tweak/determine the actuation pressure are: the difference between the internal diameter of the elastic membrane and external diameter of the container $(R - R_i)$, thickness of membrane $t$, and elasticity of the membrane. In order to investigate the effects of $t$, $(R - R_i)$, and $R$, we conducted three sets of triple experiments. In total, we fabricated 27 micro-dispensers of different configurations for studying the effect of $(R - R_i)$ and $t$ and $R$. We fabricated these micro-dispensers using 1 ml (Ø 6.6 mm) and 2.5 ml (Ø 10.1 mm) standard syringes (purchased from BD Plastipak™). We used a 1.2 mm drill bit to pierce the outlet aperture on the syringe barrel and a scalpel to cut out finger flanges of [on] the syringe barrel. Next, we inserted the piston of the syringe, pushed it inwards, and sealed the barrel of the syringe. Thus, the dispensers contain ~0.6 ml and ~2.4 ml for those

fabricated from 1 ml and 2.5 ml syringes, respectively. After filling each dispenser with ~90% di-water, we sheath them into elastic membranes previously cut to seal the aperture. All the membranes used were cut from latex tubing purchased from Kent Elastomer Products Inc. Note that all experiments have been conducted in the same manner with air occupying ~10% of the dispenser volume in order to study the influences of the presence of air on the performance of the micro-dispensers. The micro-dispensers were connected to the pressure controller using Luer-to-Luer connectors. The experiments were conducted by gradually increasing the input pressure and monitoring the membrane behavior using a desktop magnifier. After injection of air sufficient to reach the critical opening pressure for the membrane, more air has to be continuously injected to keep the needed pressure for continuous flow. At the point where the pressure inside is higher than what the membrane can tolerate, the micro-dispenser will dispense. Figure 1 shows the results of three set of experiments with membranes of the same material illustrating the effects of $(R - R_i)$, $t$, and $R$. These results can be compared with Table 3 that represents the simulation results for different cases. In general, in case of a thicker membrane or tighter sealing more pressure is required for actuating the micro-dispenser as verified by simulation results in Table 3. Also, the larger the internal diameter of the dispensers the lower the pressure for actuation if the fitting is similarly tight.

**Storage characteristics.** The permeation of gases through the micro-dispenser correlates with the penetrant and selectivity of materials of micro-dispenser components. In essence, the micro-dispenser can be manufactured with any biologically inert materials that offer low to excellent impermeability such as aluminum and glass and polymers that are widely used in food packaging industry[39,40]. The permeability data and material properties available at membrane and glove industry can be used for choosing an appropriate membrane[41–46]. In addition, the micro-dispenser can be coated in order to reduce the penetration rate through the micro-dispenser[40]. The different material preference allows to manufacture application-specific, temperature-specific and/or country-specific micro-dispensers in order to provide a customizable cost-effective product. Here, we use glass as the micro-dispenser flask material, assume that main permeation occurs through the membrane, and investigate the permeation through the membrane which may often have higher permeability rate. We tested the long-term storage capability by conducting accelerated life tests for micro-dispensers having a 1 mm diameter aperture and using neoprene as a membrane. We incubated micro-dispensers filled with di-water at 65 °C for 14 days. For calculation of the equivalent real time, we used Arrhenius equation and the activation energy for permeation reported in the literature[47–49]. We also incubated samples similar to those kept in the oven at room temperature for 180 days and weighed them on a daily basis. The results show weight loss <0.1% which conforms with literature data we used in the Arrhenius equation. Figure 2 shows that the average weight loss of seven micro-dispenser samples maintained at 14 days at 65 °C, i.e., equivalent to over 3 years at 23 °C is 0.37%. We also simulated a shelf-life test for ethanol 70% as an example of a volatile substance. For calculation of the equivalent real time, we used Arrhenius equation and experimentally measured the accelerated aging rate (i.e., temperature coefficient) to calculate accelerated aging time duration. For calculation of temperature coefficient, we maintained three different samples at 30, 40, 50, and 60 °C and compared their corresponding permeation rates. The simulated results show less than 0.1% weight loss for the simulated period which equals to more than 1 year at 27 °C. We evaluated our simulated results by maintaining 3 micro-dispenser samples at room temperature for a period of 35 days and the results confirmed our simulated shelf-life test.

| Table 3 The critical pressure simulated for different flask and membrane radii | | | | |
|---|---|---|---|---|
| Inner radius $R_i$ (mm) | Thickness $t$ (mm) | Tube radius $R$ (mm) | Critical pressure $p_0$ (kPa) | |
| | | | Case 1 | Case 2 |
| 4.8 | 2.4 | 3.3 | 146 | 140 |
| 4.8 | 1.6 | 3.3 | 118 | 114 |
| 4.8 | 0.8 | 3.3 | 74 | 73 |
| 2.4 | 0.8 | 3.3 | 246 | 239 |
| 1.6 | 0.8 | 3.3 | 346 | 339 |
| 3.9 | 1.6 | 5.0 | 67 | 65 |
| 3.9 | 0.8 | 5.0 | 35 | 39 |

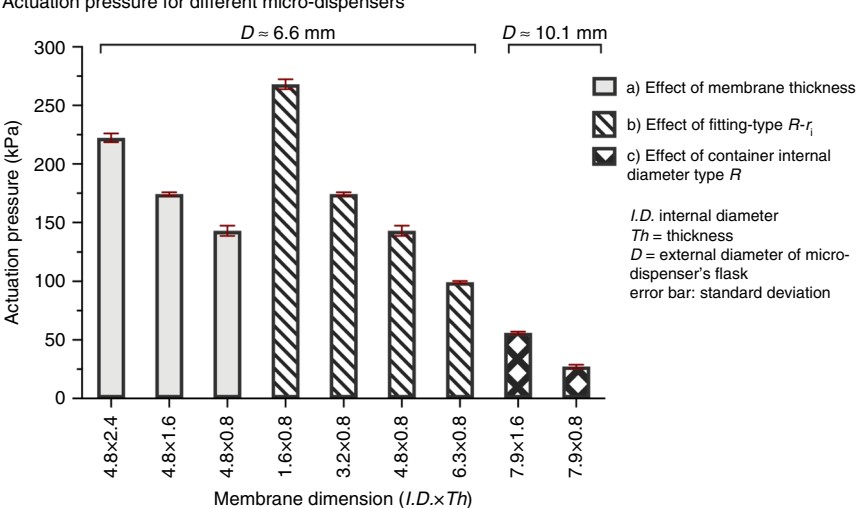

**Fig. 1** Characterization of actuation pressure for a micro-dispenser (membrane made of latex) set. **a** A dispenser of outer diameter $D = 6.6$ mm is sheathed in membranes of fixed internal diameter and different thickness, **b** a dispenser of outer diameter $D = 6.6$ mm is sheathed in membranes of fixed thickness and different internal diameters, and **c** a dispenser of outer diameter $D = 10.1$ mm is sheathed in membranes of fixed internal diameter and different thickness

**Micro-dispenser in lab-on-discs**. In order to illustrate the performance of the micro-dispenser, we demonstrate its utility in two common types of microfluidics devices, i.e., LOD and LOC. In addition, we introduce a method for clinical assays based on the micro-dispenser technique, which we term interlocking dispensing action (IDA), exemplified here by performing plasma separation from whole blood. We begin with demonstrating an example of using a micro-dispenser for sample preparation on a LOD. For an illustration of a typical sample preparation task, we demonstrate here the separation of plasma from blood cells which is the first step in many clinical assays[19]. In Fig. 3, we show

Micro-dispenser shelf-life when filled with di-water

Fig. 2 The shelf-life of micro-dispenser; the shelf-life of 7 different micro-dispensers containing an average volume of 475 μl di-water, where the flask is made of glass and the membrane of neoprene, the error bars show the standard deviation of the mean and coefficient of variation values

images of a micro-dispenser inserted in a compartment on a LOD platform covered by a piece of tape. A tacky adhesive was used to lock the micro-dispenser to the lab-disc platform as seen in the figure. The micro-dispenser is bent after the plasma microchannel and is made hydrophobic that avoids the plasma flow through blood cells microchannel. This also shows that the micro-dispenser can be positioned in various ways which facilitates the guidance of dispensed liquids via micro-dispenser on the disc. The micro-dispenser used in this experiment has two apertures covered with two membranes as seen in the inset of Fig. 3. The membranes are pre-stressed such that they do not ever open simultaneously, allowing selective and sequential release of pre-defined amounts of liquid. Before using blood samples, we filled the micro-dispensers with dyed di-water to investigate the dispensing frequency of each outlets, for details see Supplementary Figure 3. Due to the differentiation in membrane properties at a given rotational speed, the first membrane, i.e., the membrane closer to disc center stretches and liquid (i.e., here blood plasma) is dispensed. Note that the first aperture is located at a point above the level where plasma is separated from blood cells. After this the membrane reverts back to its previous state and stops the flow due to changes in liquid level in the micro-dispenser $\Delta z$. We then increase the rotational speed and gradually dispense all of the remaining blood plasma which is received in a separate chamber for further analysis. Increasing the spinning speed beyond this stretches the second membrane and allows for blood cells to flow through the second aperture. Like for the plasma, the blood cells can be transferred to a separate chamber for further analysis by increasing the rotational speed adequately. Note that there will be a residual of the blood cells that its volume depends on the position of the second aperture.

**Micro-dispenser in lab-on-a-chip devices**. In resource-limited settings, POC devices should be self-contained, run at low power, function in extreme meteorological conditions (extreme point of care or EPOC), and ideally meet the ASSURED criteria[35,50–52]. An attractive POC test device that conforms with ASSURED

Separation of blood components on a lab-on-a-disc platform

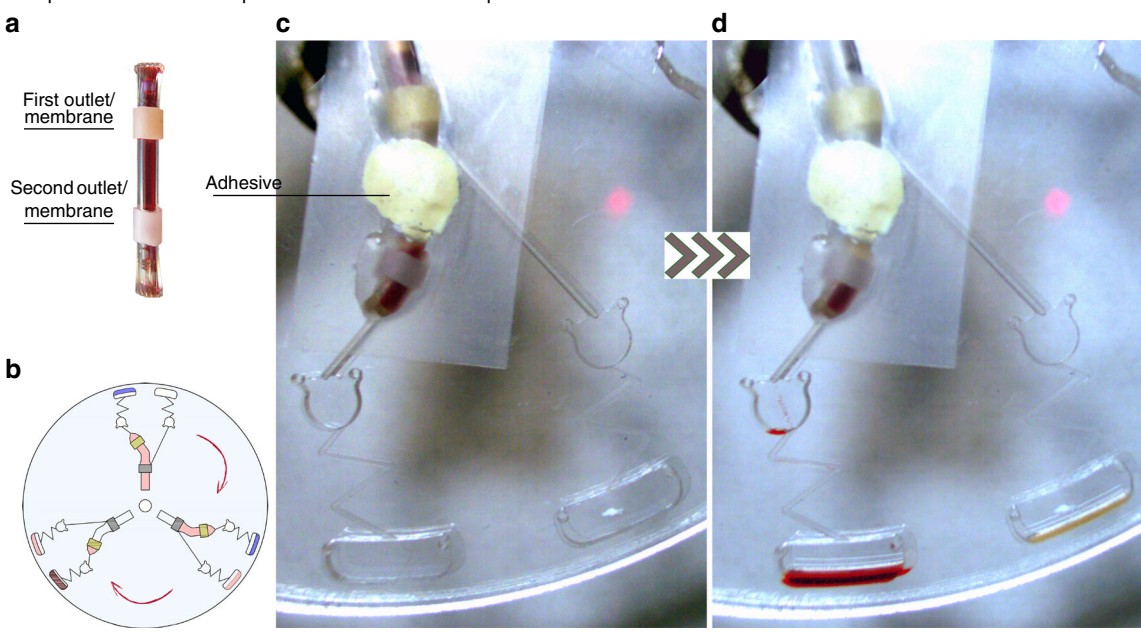

Fig. 3 Images of a rotating lab-on-a-disc with micro-dispenser insert enabling separation of blood components. **a** Micro-dispenser with two apertures covered with C-flex and latex membranes, **b** schematic of the lab-on-a-disc platform used in the experiment, **c** before actuation of the micro-dispenser, **d** dispensing blood plasma and blood cells to two different destination chambers, at different rotational frequencies due to the difference between the membrane properties of the micro-dispenser

Micro-dispenser incorporated in a lab-on-a-chip specimen

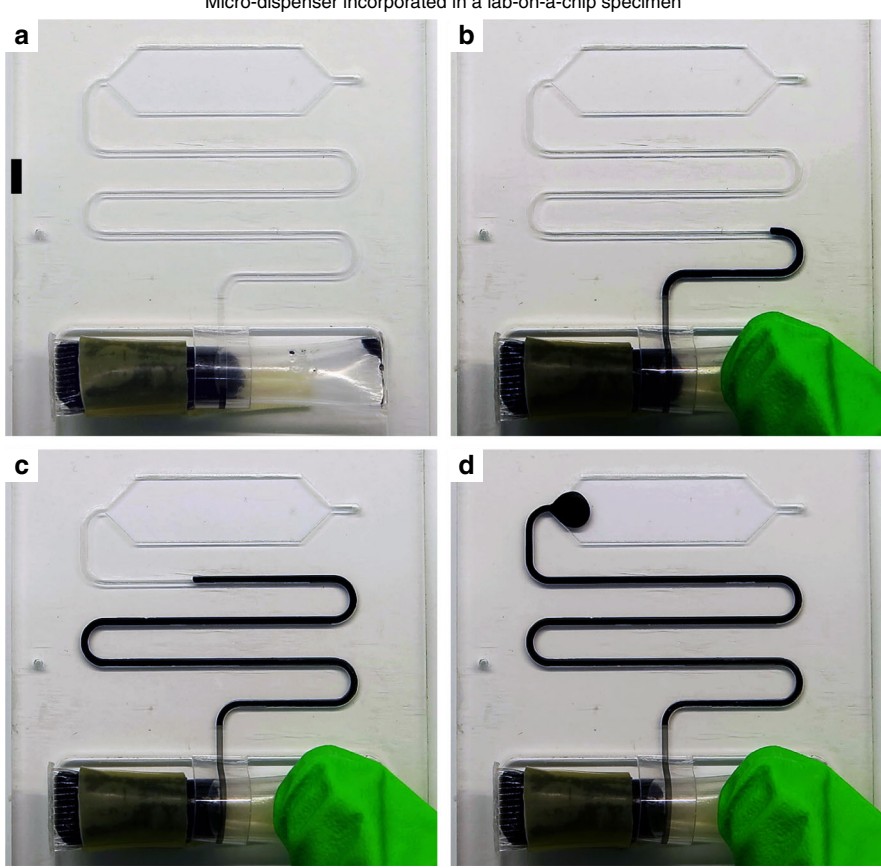

**Fig. 4** Micro-dispenser for integrating with lab-on-a-chip devices. **a** A micro-dispenser connected to a micro-structure, the scale bar is 500 μm, **b** actuating micro-dispenser by applying finger pressure, **c** applying further pressure, **d** stop applying pressure

criteria must integrate micro-dispensing and liquid storage. The attempts at developing an integrated micro-dispenser-liquid storage for LOC devices started more than a decade ago, i.e., details can be found in different review articles[53,54]. But a single generic low-cost device that is able to solve the problem of liquid storage and dispensing have never been presented. The micro-dispenser introduced here enables the integration of liquid storage and dispensing at a low-cost and is compatible with various techniques. As a proof of principle, here we integrate a micro-dispenser made from a flexible impermeable material with a LOC specimen made of poly(methyl methacrylate), see Fig. 4. The figure shows a micro-dispenser made of a flexible tubing which is half-filled with black died di-water. A given amount of liquids is pumped to the micro-structure each time the micro-dispenser is actuated, i.e., using finger pressure.

**Interlocking dispensing action (IDA).** In general, an IDA can consist of several micro-dispensers that are inserted and locked into each other. Different reagents can be pre-stored inside of each dispenser and the internal surfaces of the dispensers can be specifically modified. In the simplest configuration, an IDA consists of a micro-dispenser which is inserted in a larger receiving vessel. This configuration can be used for automating blood components separation using conventional centrifuge machines. The conventional blood plasma method in many laboratories is carried out by skilled operators that carefully aspirate the supernatant fluid, i.e., plasma or serum after centrifuging whole blood according to standard protocols. In Fig. 5, we show an IDA that consists of a dispenser that is inserted

into a standard laboratory safe-lock collection tube. The micro-dispenser is also fabricated from standard laboratory safe-lock Eppendorf tubes. The micro-dispenser is filled with blood, the IDA is placed into a centrifuge machine, and centrifuged according to the standard blood plasma separation protocols. The amount of plasma that can be collected is determined by the position of aperture on the inserted micro-dispenser. For example, assuming 50% of whole blood consisted of plasma, if the whole position is at 40% of the tube height, only 80% of plasma is collectable. Note that, as we explained before, the magnitude of centrifugal force applied to the blood volume is related to the blood plug length in tube, which allows dispensing a given volume of blood at a given relative centrifugal force (RCF). Figure 5 shows the extraction of 45% of blood plasma from 700 μl whole blood using the RCF required to dispense the entire plasma located above the micro-dispenser aperture. In order to investigate the purity of the plasma collected, we used a hemocytometer chip and a spectrometer machine to measure the absorbance of red blood cells. We compared the spectrometer and hemocytometer results for the plasma collected using IDA with the plasma collected using the conventional centrifugation method. For three samples the hemocytometer results did not show any cells in the plasma collected using IDA, see Supplementary Figure 2. The absorbance of red blood cells in the plasma sample using 575 nm wavelength shows that the blood plasma collected by IDA has 10% lower deviation from the blank control. Note that for the conventional centrifugation method we used Eppendorf Centrifuge 5810R machine and spun the sample at 900×g for 10 min. We used the same setting for blood plasma separation using IDA but we increased the speed to 1400×g for approximately 2 min to

Separation of plasma from blood using interlocking dispensing action

**Fig. 5** A configuration of interlocking dispensing analyzer assembled to automate blood plasma separation in laboratories. Interlocking dispensing analyzer for blood plasma separation is constituted of a micro-dispenser made of a safe-lock Eppendorf tube that is interlocked into a larger safe-lock collection tube. **a** An interlocking dispensing analyzer before inserting in centrifuge machine, **b** after centrifugation according to the standard protocols for 20 min, **c** blood cells in the micro-dispenser and blood plasma in a safe-lock Eppendorf tube

stretch the membrane and dispense the plasma into a separate tube. Using the IDA concept, the entire plasma separation process is automatized and can also be integrated with further downstream analysis.

## Discussion

The micro-dispenser we propose is a device that stores and dispenses different liquids across different microfluidics platforms. In order to ensure that the micro-dispenser can be used for dispensing various liquids, we investigated its capability of dispensing liquid with different properties. We used different sets of micro-dispensers filled with di-water, methanol, and solutions of 75% glycerol and 25% di-water. For these specific micro-dispensers, our results show that the viscosity has a negligible effect, while the micro-dispenser filled with denser liquids need lower force for actuating in LOD platforms.

In LOD platforms, two additional parameters contribute in a more sensitive actuation and precise liquid dispensing. These are the initial amount of air inside the micro-dispenser and the liquid plug length, which both counter the effect of centrifugal pressure. The initial amount of air gradually expands after each dispensing which is due to dispenser's flask is made of rigid materials. The expansion of air develops a negative pressure (partial vacuum), that will affect the dispensing mechanism. Also, in these platforms at the constant rotational frequency, the centrifugal pressure applied decreases as the liquid plug length decreases[55]. This allows for dispensing only a given amount of liquid at a given rotational frequency and to dispense the same volume of liquid again we need to increase the rotational frequency. The magnitude of this increase in centrifugal pressure relates to the amount of gas inside the micro-dispenser, and the length of liquid plug[56]. In general, the larger the gas volume, the lower the negative pressure developed, and the shorter liquid plug, the lower centrifugal pressure applied. However, when the micro-dispenser's flask is rigid and the initial amount of air is quite low, a dead liquid volume may be considered that can be calculated when the standard air volume and liquid volume are known. In LOC platforms, i.e., when actuated using finger pressure, no negative pressure is developed as the flask of the micro-dispenser is made or partly made of a flexible material that shrinks in response to the air pressure decrease. Therefore, the volume of liquid that can be dispensed directly correlates with the amount of air. Hence, as a rule of thumb, the same amount of air must be available to dispense the entire liquid using finger pressure. In this case, the accuracy of dispensing can be improved by indirectly pressurizing the micro-dispense using a screw-based

system that provides more control in applying pressure. Also, modifying the micro-dispenser design, e.g., using the same mechanisms used for volumetric dispensing pipette, can be an alternative approach[57,58]. The micro-dispenser can also be actuated using a piston-based mechanism, which can provide more accurate dispensing.

In conclusion, several scientists and engineers have suggested that LOC technologies require a generic proportional valve combined with a long-term liquid storage chamber that is very low-cost, very simple, scalable, and broadly compatible with different microchips[2,32–37,50,59–61]. This micro-component would then operate basically just as a dispenser does in centralized laboratories[2,32–37,50]. In this paper, we reported exactly such a micro-dispenser device that enables long-term on-chip liquid storage and dispensing of reagents. It is a low cost, accurate, adjustable, robust, and multifunctional device that is broadly compatible across different fluidic platforms. We have shown its dispensing accuracy both for LOC and LOD platforms and compared our results with the standard micro-dispensers. We demonstrated its shelf-life for di-water and ethanol 70% over an accelerated age time duration of 2 years and 1 year, respectively. We presented the utility of micro-dispenser in sample preparation in LOD platforms and introduced a robust and simple method for the automation of clinical processes based on the micro-dispenser technology. The micro-dispenser technology intends to enable much-needed affordable POC diagnostic services, especially at resource-limited settings. We also believe that this technology is not limited to applications shown in this article but it may be the tool that accelerates the growth of the microfluidics industry and finally moves more fluidic platforms from the academy to the open market.

## Methods

**Micro-dispenser mechanism**. In Fig. 6, we show a simple configuration of the proposed micro-dispenser and two different approaches for actuating the micro-dispenser. Here, the micro-dispenser comprises a tube with an aperture and an elastic membrane that covers the aperture. The micro-dispenser is actuated when its internal pressure becomes greater than the force required to stretch the membrane. At such an internal pressure the membrane stretches and provides a path for liquid to discharge, as shown in the subfigures. Figure 6c shows how manually squeezing a flexible section can actuate the micro-dispenser. This principle can be used for actuating the micro-dispenser when it is used in LOC platforms. Figure 6e shows a micro-dispenser placed on a LOD where increasing the artificial gravity by spinning actuates the dispenser.

**Fabrication**. The simplicity of the proposed micro-dispenser enables various methods of fabrication both for prototyping and in mass production. Figure 7a–d shows a method for fabricating a cylindrical micro-dispenser that is more suitable for producing prototypes at research centers and universities. The figure shows the

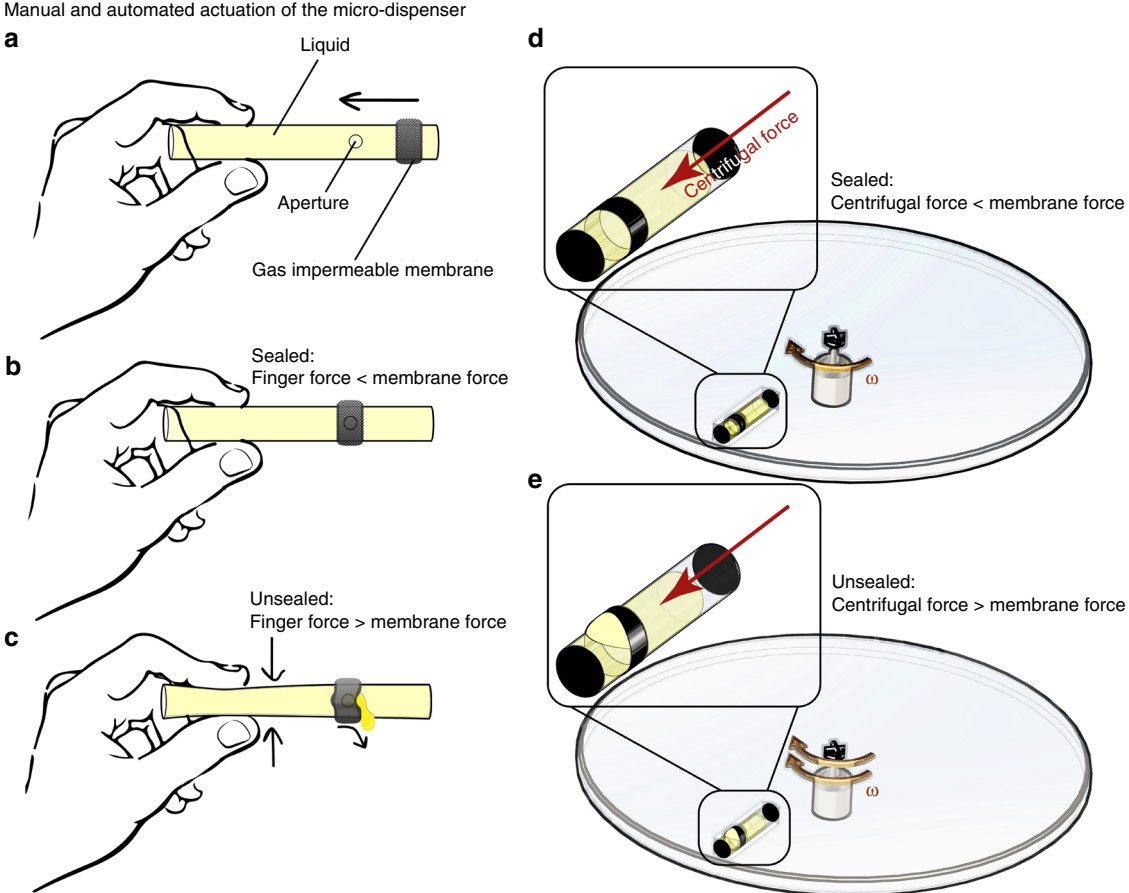

**Fig. 6** Schematic illustration of the micro-dispenser technology (two configurations). Panels (**a**) and (**b**) show the micro-dispenser components and assembly, panel (**c**) shows manual actuation by pressing the micro-dispenser body. Panel (**d**) shows the micro-dispenser inserted in a lab-on-a-disc platform, when the centrifugal pressure is lower than membrane resistance, panel (**e**) by increasing the spinning speed, centrifugal force stretches the membrane that allows a temporary release of liquid. The micro-dispenser size is exaggerated for clarity

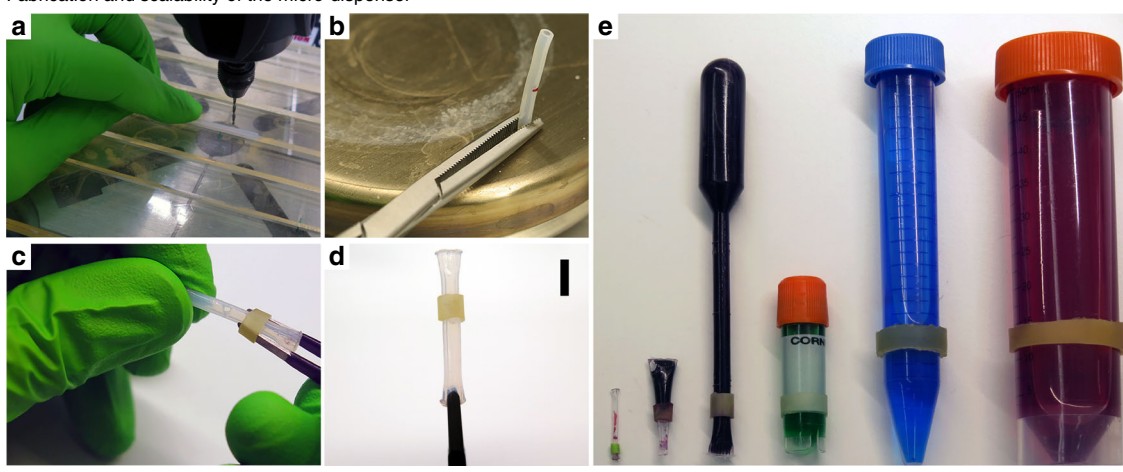

**Fig. 7** Scalability and procedure of producing a cylindrical micro-dispenser. **a** Drilling the outlet aperture on a predefined length of an impermeable tube, **b** sealing each side of the tube, **c** sheathe in a predefined length of an elastic membrane, **d** close-up view of a micro-dispenser, the scale bar is 500 µm, **e** examples of scalability of micro-dispensers

components required for fabricating the device, i.e., an impermeable tube and a flexible membrane with low permeability and preferably high elasticity. The tube is measured, cut, and a drill is used to pierce one or multiple apertures. Both sides of the tube are sealed using a clamp and a hot plate. The tube is sheathed in the membrane, which was previously measured and cut by a sharp scalpel, using a three-pronged plier. At this point, the membrane is half-covering the aperture and

the micro-dispenser is autoclaved and packed, if necessary. Before use, the reagent is dispensed into the tube and the membrane is pulled to now completely cover the aperture. Note that, the micro-dispenser can be filled using a syringe or an Eppendorf micro-loader depending on the micro-dispenser dimensions. All components of the micro-dispenser are purchased from well-known suppliers that allow for estimating the fabrication cost of prototype micro-dispensers. Depending

on the materials used the micro-dispenser costs approximately 2–6\$, i.e., considering the average labor wage of the European Union. For other configurations of micro-dispensers, e.g., when we use safe-lock Eppendorf tubes or pipettes, prototyping costs will be slightly higher or lower depending on the material used for fabrication. For mass production, the micro-dispenser flask can be produced in bulk using different standard mass production methods, e.g., injection molding for polymeric materials, molding for metals such as aluminum. The membrane raw material can be purchased from mass product suppliers and cut into precisely measured pieces using a robotic knife. The robotic knife can be connected to holders and conveyors that hold, transfer and measure the membrane for cutting. The micro-dispenser can be assembled by developing customized robotic arms that enable exact positioning of the membrane.

In Fig. 7e, we show how conventional tubing, pipette, pipette tips, and storage tubes can all be used to manufacture micro-dispensers in various sizes. Note that micro-dispensers can be manufactured in different shapes as well. However, we only show cylindrical shapes because of the abundant availability of cylindrical vessels. The smaller micro-dispensers shown in subfigures can be used directly or with slight modification as inserts in LOC and LOD platforms. The bigger micro-dispenser made from a pipette can be used in several different settings when an airtight dispenser that prevents air exposure to the liquid inside is needed. For example, this is highly desirable in medical dispensers such as eye and ear droppers.

**Analytical treatment**. An exact analysis of the micro-dispenser functioning, from a mechanical viewpoint, is complicated and perhaps of somewhat limited ultimate relevance. However, a qualitative analytical treatment results in an approximate expression that elucidates the micro-dispenser functionality. Figure 8 shows a schematic of a micro-dispenser inserted in a disc, spinning with an angular velocity $\omega$. The rotation center is situated at a distance of $H$ from the distal end of the micro-dispenser, whereas the aperture is situated at $h$. The current liquid level in the micro-dispenser is denoted by $z$, the fluid level at the start is $z_0$ and the process ends at $z = z_1 < z_0$. This means that the total liquid volume released is $V = A(z_0 - z_1)$, with $A$ the inner tube area. The inset figure shows a section through the tube and the membrane, where the unstressed membrane has an inner radius $R_i$ and an outer radius $R_o = R_i + t$, with $t$ its unstressed thickness. The membrane is initially stretched, so that the deformed inner radius $r_i = R$, and releases liquid, at once the membrane is stretched beyond this, $r_i > R$.

The mechanical analysis is based on at least four assumptions and simplifications. First, we assume that flow starts and continues when the pressure $p$ at $z = h$ is above a critical pressure $p_0$, which depends on the membrane properties, but considers the tube as completely rigid. This means that the membrane opens the aperture, when the pressure from the inside is such that $R_i > R$ for the whole region between the aperture and one membrane edge. Second, we assume that the sectional area of the tube is small so the tube can be seen as thin around its axis. A third assumption is that there is no under-pressure developing within the tube during fluid outflow. We finally assume that the friction between tube and membrane can be neglected. Based on these simplifications, the critical pressure

$p_0$ can be approximately calculated from a plane-strain model of the circular membrane, seeing it as infinitely long, but considering the length change related to the initial membrane radial expansion (but this is not further considered here). The critical pressure at the aperture can be equated to the pressure from the fluid of density $\rho$ at an angular velocity $\omega$ of the rotating tube. This can be calculated from an integral of the centrifugal forces from the fluid surface level $z$ to the aperture level $h$, which based on the geometry in Fig. 1 leads to a pressure from rotation as:

$$p(z) = \frac{1}{2}\rho\omega^2\left[(H-h)^2 - (H-z)^2\right] \tag{1}$$

From experimental inspections, the membrane first locally releases from the container when critical pressure is reached, and this release immediately spreads around the cylinder circumference. Depending on the setting (static or rotational), the release of at least one side of the membrane from the cylinder occurs soon thereafter, and this release allows fluid flow. Equating this pressure to the critical $p_0$ allows, when $H$, $h$, $\rho$, and the initial height $z_0$ are known, the calculation of the angular velocity $\omega$ at which the membrane first opens $p(z_0) = p_0$. With a prescribed higher angular velocity $\omega$, it also allows the calculation of the filling height $z_1$ at which outflow ends, $p(z_1) = p_0$. For an improved analysis of the critical pressure, with a finite length membrane and prescribed position of the aperture, we also simulated the membrane response using COMSOL Multiphysics (version 5.3, COMSOL AB, Stockholm, Sweden)[62]. Drawing on the experimental observations, we used a 2-D axi-symmetric simulation, with known geometric and material parameters, and simulated a uniform radial expansion for the membrane inner radius, until the membrane reached the tube radius. At this state, the highest pressure shown between the aperture and the nearest membrane edge was considered as the critical pressure. As few examples, Table 1 lists the critical pressures for membranes made of an assumed incompressible Neo-Hookean material with a constitutive constant $\mu = 400$ kPa, and different inner radii $R_i$ and thicknesses $t$, for two different tube radii $R$. Case 1 is related to a membrane with unstressed length 4 mm, and a central aperture, while Case 2 used a 3 mm membrane, with an aperture 0.5 mm from one of its ends, see Fig. 8b. The results listed in Table 3 can be compared to experimental results in Fig. 1. Considering the rough material model, the neglection of friction between membrane and tube and the imperfections in the experiments, the correlation must be considered as rather good, indicating that the simulation model can be used for predictions of the response in different designs, when more accurate data are available. It is noted that the simulations consider neither gravity nor centrifugal pressure, but only sees an air over-pressure.

## Data availability

The data that support the findings of this study are available from the authors upon reasonable request, see author contributions for specific datasets. Full input specifications for performed simulations in COMSOL Multiphysics are available after request to author A.E.

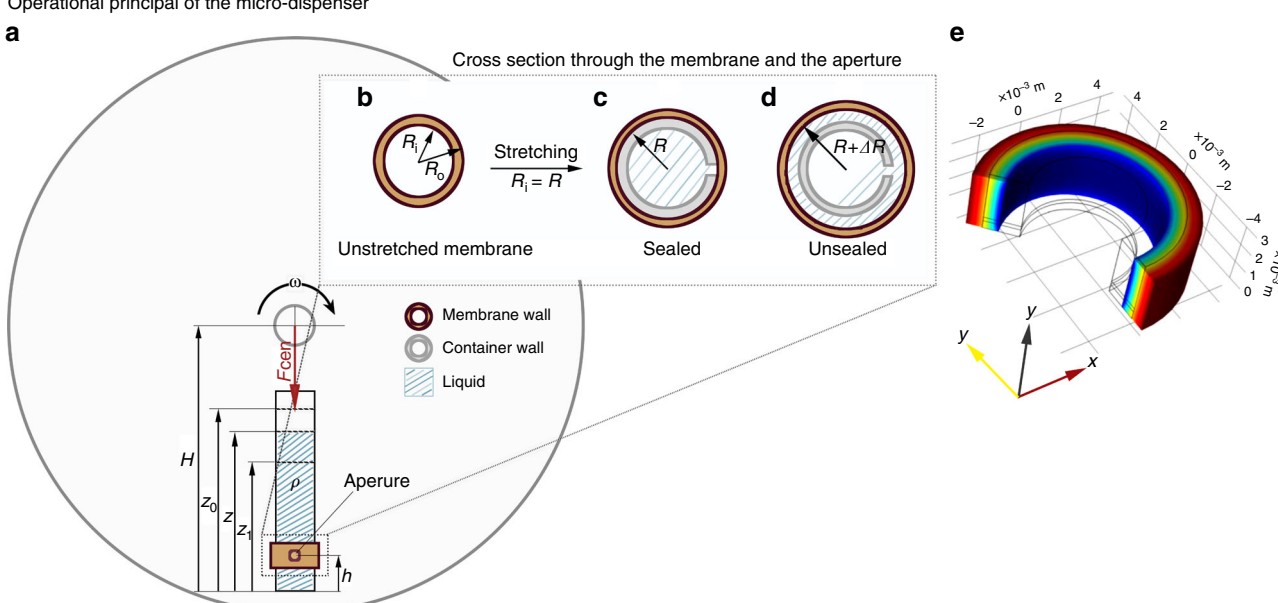

**Fig. 8** The analytical treatment of the micro-dispenser actuated by centrifugation. **a** Schematic of the micro-dispenser inserted in a rotating frame together with its actuation mechanism. The inset shows a section through the membrane and the tube. **b** The membrane un-stretched. **c** A tube (of outer radius $R$) is sheathed in the stretched membrane, providing a hermetic sealing. **d** At given speed, the membrane is stretched to inner radius $R_i > R$, and liquid is released. **e** Example results from simulations, showing deformed membrane and color-coded radial pressure for an opening membrane of $R_i = 4.8$ mm, $t = 1.6$ mm on a tube with $R = 3.3$ mm, Case 2

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

## Acknowledgements

The authors thank Robert Bouwens, Julie Bouwens, and Jahanshah Christu for their support and helpful suggestions throughout the research, Esmail Pishbin and Noa Lapins for their kind cooperation, and Andreas Lundström, and Frida Nilsson for their assistance in illustrating of figures.

## Author contributions

A.K. performed the experiments. A.K., A.E., M.M., and A.R. analyzed the experimental results. A.E. performed the simulation and provided the theoretical section. A.R. and M.M. supervised the project. All authors contributed to writing the manuscript.

## Additional information

**Competing interests:** A.K. and POIETAI L.L.C. hold commercial rights including but not only a patent pending (PCT International Patent Application No. PCT/US17/22956) to the micro-dispenser and micro-dispenser-based products. The other authors declare no competing interests.

