## [Peer Review File · Nature Communications]

Reviewers' comments:

Reviewer #1 (Remarks to the Author):

The manuscript, entitled "A micro-pipette for long-term storage and controlled release of liquids" by Kazemzadeh et al., presents a device for dispensing fixed volumes of reagents in lab-on-a-disc platforms with an extended shelf-life. The micro-pipette consists in a flask having an aperture that is sealed with an elastomeric membrane. Pressure or centrifugal force applied to the flask forces liquid out of the flask by displacing a sealing membrane.

The integration and release of reagents in portable diagnostic devices is critical for the good functioning of such devices. But I feel these issues have been largely solved. Many point-of-care diagnostic products are available on the market with reagents integrated and with long shelf lifetime (2 years in some cases and 18 months most of the time). Storage of reagents has been done with reagents both in dry and liquid form. Lab-on-a-disc products from Biosurfit are just one example. Controlled release has also been done using various spinning rates and geometries using, for example, burst valves or hydrophobic barriers. Another major criticism is that the manuscript is extremely superficial and ignores showing reasonable applications, and "jumps" from controlled release of liquids to blood aliquoting. There is no coherence in the work and presentation of what has been done. The text is hard to follow and lacks technical details. I do not understand what is shown in figure S1, for example. The abstract sells the work without explaining what was done. In summary, this manuscript is not suitable for publication in Nature Communications or another journal.

Reviewer #2 (Remarks to the Author):

The article "A micro-pipette for long-term storage and controlled release of liquids" describes for the first time a concluded reservoir for long-term storage of reagents, with integrated, pressure operated, normally closed, passive check-valve for dispensing. Aliquots of the reagent can be repeatedly released from the reservoir, in proportion to the applied pressure. Pressure for operation of the release valve can be built up by centrifugal force (hydrostatic pressure) or by mechanical actuation (not demonstrated). Multiple valves with different release pressure can be implemented at different positions of the reservoir to release different fractions of a liquid at different times. In combination with a centrifugal device, this allows to implement more complex multiple fluidic operations, e.g. the fractionation of blood into cells and plasma, as well as the downstream

collection of these fractions. Application of the reagent reservoir with integrated release valve was demonstrated for the centrifugal microfluidic platform, but it could also be useful in conjunction with other microfluidic platforms, dispensing applications or drug delivery.

The presented work provides novel and original aspects. It discloses a concept which is of significant interest to the liquid handling community.

However, the suggested approach, and applications are only described on a superficial concept level, and not much quantitative information about the manufacturing and performance characterization is provided. Hence, the suitability of the micro-pipette for future applications does not become clear. Also, the state of the art is reviewed only superficially, and neglects certain significant achievements in the field. The suggested improvements are listed further down.

General comments

- The release valve of the described micro-pipette works according to the principle of the well-known original Dunlop check valve formerly used for bicycle tubes (a scheme is provided here: <https://de.wikipedia.org/wiki/Fahrradventil#Dunlopventil>), US455899A, invented in 1891.
- Line 37: Although the check-valve of the micro-pipette is a passive valve an external power source (centrifugal device, piston, manual actuation, etc.) is required for operation.
- Line 113, Fig. 2. Is the membrane (elastic tube) only locally stretched, near the aperture, or all around the reservoir? The required energy for local stretching might be less than for stretching the entire tube by ΔR .

-

Suggested improvements of high priority

- State of the art: It is the first time that a reagent reservoir with an integrated Dunlop type check valve has been published to allow both, storage and repeated proportional reagent release. However, further state of the art reagent storage and release concepts should be referenced as part of the introduction (and line 20 – 22 of the abstract should be modified accordingly):
- Line 57: I suggest to consider the following references to more thoroughly discuss the advantages of your innovation in the light of the diverse aspects of the state of the art.
 1. For aliquoting of released reagents, good results have been obtained by implementing specific downstream unit operations: (a) DOI: 10.1039/c7an00547d: A fully integrated microfluidic

platform for highly sensitive analysis of immunochemical parameters; (b) <https://www.researchgate.net/publication/309242183>); (c) Schembri, C.T. et al. (1992) Clin. Chem. 38/9, 1665-1670 Portable simultaneous multiple analyte whole-blood analyzer for point-of-care testing.

2. For sequential release of reagents from stick packages (Ref. 16) a microfluidic timer has been introduced as downstream unit operation which provides repeated release of reagents as downstream unit operation [(a) DOI: 10.1039/C4LC01269K, A microfluidic timer for timed valving and pumping in centrifugal microfluidics; (b) B. Johannsen, et al. "Disk-integrated repeated dispensing of 200 nl volumes for the automation of pyrosequencing" 2017 MicroTAS 2017, Savannah /USA, 22.-26.10.2017)]

3. For pressure driven proportional release of reagents from blisters (Ref. 17) a piston based approach is available (Microfluidic Chipshop, Blister Driver – ChipGenie® edition BD).

4. For stick-packs in another pressure driven platform dosage has been accomplished by an elastomeric downstream valve (World Academy of Science, Engineering and Technology International Journal of Physical and Mathematical Sciences Vol:7, No:8, 2013, Long-Term On-Chip Storage and Release of Liquid Reagents for Diagnostic Lab-on-a-Chip Applications D. Czurratis, et al.).

5. In addition, IQuum Inc (Allston, MA) patented the Liat Molecular Analyzer which is based on the lab-in-a-tube (Liat) technology. The Liat tube serves as reagent container. The flexible tube serves as contains all assay reagents pre-packed in tube segments, separated by peelable seal which are formed by a thermal weld of the plastic tube. By applying pressure to the tube segments adjacent to each seal, the seal can burst open to release reagents. In the Liat analyzer, multiple sample-processor modules are aligned with the Liat tube. Each module consists of an actuator and a CLAMP, whose positions can be controlled to manipulate a test sample within a tube. By synchronizing the motion of the actuators and clamps, various sample processes can be conducted within a tube. Such processes include ADJUSTING a LIQUIDS VOLUME in a segment; Stepwise RELEASING of a reagent to the adjacent segment; mixing; etc.

- Line 168: Fig. 3: The terms "Net volume" and "total volume" need to be defined. No error bars are provided for the net volume.

- Line 209, Fig. 4. The resolution of the image of the two-membrane micro-pipette is too poor, and details are hardly visible. In addition, it is not clear how the liquid is guided from the membrane outlet of the micro-pipette into the receiving channel. This should be depicted in more detail. Also, the micropipette integrated in the disk seems to be tilted, and the reason for this should be explained.

- Experimental data provided by the authors

1. Although the analytical treatment is described in the article, no comparison between theoretical and experimental data (critical pressure) is made in the results / discussion section. This should be included (see Table 1).

2. For the analytical description, the equation used to calculate the critical pressure p should be provided as well (line 117)

3. For a specified setup (volume, liquid, and membrane material) experimental characterization and statistical data evaluation should be performed more comprehensively, especially providing quantitative data (Tables). Typical qualification comprises the following: typical volume range, setting a specific target volume (nominal volume), resolution and calibration, determination of actual volume including precision, accuracy (systematic error), repeatability (random error) (inter- and intra-pipette run accuracy and CV, load-to-load accuracy and CV), influence of dead air volume (for operation by mechanical actuation), dead-liquid volume, amount of residual liquid), influence of the temperature and barometric pressure. These values should be compared to the requirement of laboratory devices (i.e. pipettes, dispensers, to which the abstract refers (see also ISO 8655 or alternative standards and quality assurance documents for qualification of pipettes).
4. For the long time storage tests, data of real-time tests should be added as far as available (those used for calibration, e.g. for the first six month). Also, more volatile liquids, such as ethanol should be tested and data provided.
5. For the blood plasma separation, the hematocrit of the sample, the separated blood volume, the separation time, the plasma yield and purity needs to be provided as well, and the values should be compared to clinical standards or reference methods.
6. A manufacturing guideline should be provided, e.g. materials of the glass tubes, filling of the glass tubes, sealing procedure, implementation of aperture,
7. The potential for large scale manufacturing should be critically discussed (part numbers, costs) and be compared to other established approaches of reagent prestorage and release.
 - Information about the use of the micro-pipette for blood plasma separation and other applications demonstrated should be added to the abstract, as well as some quantitative key characteristics of the micro-pipette.
 - Line 239: In my opinion, the unavailability of experimental and manufacturing data is not acceptable for a scientific paper. A comprehensive set of original experimental data needs to be provided or referenced, to ensure reproducibility.

Further suggestions / discussion

- Line 1: The suggested micro-pipette could also be designated as “micro-dispenser” as it can only dispense, and aspirate liquids.
- Line 26: consider: "the approach is INTENDED to [...]"
- Line 147: For details about the experimental procedure the electronic supplement should be referenced here.

- Line 164: A comparison of the shelf-life between two systems appears only meaningful if two specific systems are compared. Both systems, the micro-pipettes, and the reference system could be probably produced from various materials.
- All error bars should be defined in the corresponding figure legends.

Reviewer #3 (Remarks to the Author):

It is certainly true that the two drawbacks of POC devices for field use addressed in this manuscript are relevant, and among the reasons why these devices are not commonly used to date. In this respect, the work presented is very relevant. This work builds well on previous work looking at other means to achieve liquid control and storage (e.g. blister packages) and is novel in its approach. The work would be more valuable if additional result data can be added. It is assumed that this data is not included in the confidential data mentioned. Specific points are outlined below:

The manuscript mentions that the device can be actuated with finger, centrifuge or other means. Of particular interest would be the finger activation, as this would be very useful in resource limited settings. Presumably glass cannot be used for this. The authors should provide data showing repeatability for finger actuation of polymer based devices. As one of the central themes of the work is repeatability, this is very important to show. How easy is finger actuation – authors provide flask inner radius, but not outer, and it is difficult to gauge how easy it is to use finger actuation. Very small diameter tubes can be very stiff.

The authors also mention low cost and scalability as advantages, but provide no data on this. It would be useful to know what typical cost points are for this type of pipette, and also what is meant by scalability. Is it the ability to increase volume of device, or scaled manufacture of the device?

Would it be possible to provide data in table 1 for the 1mm pipette referred to in fig 3, and also provide the volume of the 1mm pipette? It would appear that most volumes are several 10's to 100's of micro litres, and a graph showing aliquoting accuracy for the full volume of the pipette (e.g. 25ul or 475ul) is important. It is also not clear how the viscosity of different fluids (expected in real use) would influence dispensing accuracy.

The authors need to motivate in more detail why their accelerated life time tests are valid in this device? Material used for the test (glass) is ideal for storage, but is not always favoured in resource limited settings, so that data for polymer materials is required. Even though tests were done, this data is not presented. Additionally, tests are done with de-I water, and data for expected field reagents would be useful as these are often corrosive in nature.

Minor points to be addressed:

Line 54: till should be until (till is seldom used)

65: on-chip is not correct terminology for lateral flow. I would suggest on-test or leave out

108: delta r should be delta R

Figure 2: figure shows R as outer radius (appears so from schematic). This appears incorrect.

148: Nano = nano

153: significantly improved – can this be quantified?

204,208: number of places have e.g. and then etc. I would suggest removing the etc in all cases.

References need reworking as are not all in same format.

General comments:

Work is relevant and I believe novel. It would be difficult to reproduce given the detail provided. However, if authors can make the suggested changes, then it is certainly work which will be very valuable to the community of people developing point of care diagnostics, which are critically needed.

REVIEWER REPORT and response

Below, you can find the original comments and suggestions given by each reviewer and our responses. For convivence, we have classified and addressed them accordingly. The original reviewer's comments are colored and underlined. We provide general response followed by detailed response with reference to the article. For convenience, we have provided the line number for all new information and have embedded newly added and modified figures and tables.

Reviewer #1

Comments:

The manuscript, entitled "A micro-pipette for long-term storage and controlled release of liquids" by Kazemzadeh et al., presents a device for dispensing fixed volumes of reagents in lab-on-a-disc platforms with an extended shelf-life. The micro-pipette consists in a flask having an aperture that is sealed with an elastomeric membrane. Pressure or centrifugal force applied to the flask forces liquid out of the flask by displacing a sealing membrane. The integration and release of reagents in portable diagnostic devices is critical for the good functioning of such devices. But I feel these issues have been largely solved. Many point-of-care diagnostics products are available on the market with reagents integrated and with long shelf lifetime (2 years in some cases and 18 months most of the time). Storage of reagents has been done with reagents both in dry and liquid form. Lab-on-a-disc products from Biosurfit are just one example. Controlled release has also been done using various spinning rates and geometries using, for example, burst valves or hydrophobic barriers. Another major criticism is that the manuscript is extremely superficial and ignores showing reasonable applications, and "jumps" from controlled release of liquids to blood aliquoting. There is no coherence in the work and presentation of what has been done. The text is hard to follow and lacks technical details. I do not understand what is shown in figure S1, for example. The abstract sells the work without explaining what was done. In summary, this manuscript is not suitable for publication in Nature Communications or another journal.

Response:

Dear reviewer, we appreciate your comments and believe that we have addressed your concerns by implementing your suggestions as well as the other reviewer's. As a result, our manuscript has become far better and so much improved. In general, we agree that the previously submitted manuscript was superficial in many aspects and we had not clarified the advantages and utility of the micro-dispenser coherently. We also agree that the integrated release of reagents has been addressed in some LOC devices. However, we must emphasize that these solutions have not integrated the liquid storage with dispensing into a single generic device. Besides, the majority of previously developed technics have been product specific, sophisticated and complex liquid handling methods that have limited their applicability compatibility with different microfluidics platforms.

Reading your feedback, we have taken the opportunity to clarify the novel and original aspects of our liquid storage and dispensing concept. We have modified the entire manuscript, added new figures and tables and updated the old table and figures by adding more experimental data. The revised manuscript has become coherent and the micro-dispenser contribution in growth of the microfluidics industry has been thoroughly discussed. As your comments have been reflected more specifically in other reviewers' comments, we would like to respectfully bring your attention to our responses to the other reviewers, where we have addressed the comments point by point and in details. Briefly, except previously known figure 1 we have changed, modified or updated all previously submitted figures and tables and added two new tables and four figures. We have extended the introduction section and thoroughly discussed the state of art technologies in liquid handling and liquid storage in microfluidics platforms. In method section, we have provided a detailed manufacturing guideline for specific micro-dispensers, discussed about a mass-production approach and the manufacturing costs. The results section now consists of more experimental results that verify our analytical simulation approach. It also represents our newly added characterization results that can be used as a general guideline for design and development of specific micro-dispensers both in lab-on-a-chip and lab-on-a-disc devices. We have shown more application instances by integrating a pressure-driven micro-dispenser with a specimen lab-on-a-chip device and introducing a novel method for automating clinical procedures that we call interlocking dispensing action.

Reviewer #2

Comments: overview:

The article “A micro-pipette for long-term storage and controlled release of liquids” describes for the first time a concluded reservoir for long-term storage of reagents, with integrated, pressure operated, normally closed, passive check-valve for dispensing. Aliquots of the reagent can be repeatedly released from the reservoir, in proportion to the applied pressure. Pressure for operation of the release valve can be built up by centrifugal force (hydrostatic pressure) or by mechanical actuation (not demonstrated). Multiple valves with different release pressure can be implemented at different positions of the reservoir to release different fractions of a liquid at different times. In combination with a centrifugal device, this allows to implement more complex multiple fluidic operations, e.g. the fractionation of blood into cells and plasma, as well as the downstream collection of these fractions. Application of the reagent reservoir with integrated release valve was demonstrated for the centrifugal microfluidic platform, but it could also prove useful in conjunction with other microfluidic platforms, dispensing applications or drug delivery. The presented work provides novel and original aspects. It discloses a concept which is of significant interest to the liquid handling community. However, the suggested approach, and applications are only described on a superficial concept level, and not much quantitative information about the manufacturing and performance characterization is provided. Hence, the suitability of the micro-pipette for future applications does not become clear. Also, the state of the art is reviewed only superficial, and neglects certain significant achievements in the field. The suggested improvements are lined out further down.

General Response:

Dear Reviewer, we sincerely thank you so much for reviewing our manuscript and for providing detailed suggestions and thoughtful comments. We have revised the manuscript considering your comments and those from other reviewers. As a result, the work has been much improved and become far better. In general, we have extended our abstract and added nearly one page to our introduction in order to carefully implement your comments and suggestions. The method section has been also much improved, as we have added a detailed guideline for producing specific micro-dispensers and have discussed the fabrication costs and a mass-production method. We have revised the analytical section and added more simulation results that verify our newly conducted experimental tests. Our results and discussions section has become far better and coherent, thanks to your detailed comments. In the result section we have added two Tables and one Figure. Note that we have not only implemented your specific comments but also provided characterization results for specific micro-dispensers for pressure driven systems that was pointed out in your overview comment. We have also completely revised our application section and added two new figures. In the revised application section, we have not only implemented your comments but also have added several novel applications among them a novel method for performing centrifugal based bioassays. In order to simplify tracking the changes, we made minor changes in your thoughtfully classified

comments and addressed them accordingly. Your original comments are colored and underlined.

General Comments:

Comment 1:

Line 37: Although the check-valve of the micro-pipette is a passive valve, an external power source (centrifugal device, piston, manual actuation, etc.) is required for operation.

Response:

Previously *line 37* has been rectified and the new statement is found in **Line 114-116**. This correction is also considered throughout the paper. **Line 114-116 says:** “A robust and generic solution to abovementioned problems must be simple, low cost and easy to actuate i.e., using only forces inherent to the microfluidic platform or simple finger pressure.”

Comment 2:

Line 113, Fig. 2. Is the membrane (elastic tube) only locally stretched, near the aperture, or all around the reservoir? The required energy for local stretching might be less than for stretching the entire tube by ΔR .

Response:

Thank you for remarking this important issue in understanding the operational of the micro-dispenser. The membrane first locally releases from the cylinder when critical pressure is reached, and this release immediately spreads around the container circumference. Depending on setting (static or rotational), the release of at least one side of the membrane from the cylinder occurs soon thereafter, and this release allows fluid flow.

The explanation added to the manuscript (see **Line 234-238**) follows as: “From experimental inspections, the membrane first locally releases from the container when critical pressure is reached, and this release immediately spreads around the cylinder circumference. Depending on setting (static or rotational), the release of at least one side of the membrane from the cylinder occurs soon thereafter, and this release allows fluid flow.”

Suggested improvements of high priority:

Comment 3:

Suggestions:

a- Further state of the art reagent storage and release concepts should be referenced as part of the introduction (and line 20 – 22 of the abstract, should be modified accordingly).

b- Information about the use of the micro-pipette for blood plasma separation and other applications demonstrated should be added to the abstract, as well as some quantitative key characteristics of the micro-pipette.

c- Line 57: I suggest to consider the following references to more thoroughly discuss the advantages of your innovation in the light of the diverse aspects of the state of the art:

c1- For aliquoting of released reagents, good results have been obtained by implementing specific downstream unit operations: (a) DOI: 10.1039/c7an00547d: A fully integrated microfluidic platform for highly sensitive analysis of immunochemical parameters; (b) <https://www.researchgate.net/publication/309242183>; (c) Schembri, C.T. et al. (1992) Clin. Chem. 38/9, 1665-1670 Portable simultaneous multiple analyte whole-blood analyzer for point-of-care testing.

c2- For sequential release of reagents from stick packages (Ref. 16) a microfluidic timer has been introduced as downstream unit operation which provides repeated release of reagents as downstream unit operation [(a) DOI: 10.1039/C4LC01269K, A microfluidic timer for timed valving and pumping in centrifugal microfluidics; (b) B. Johannsen, et al. "Disk-integrated repeated dispensing of 200 nl volumes for the automation of pyrosequencing" 2017 MicroTAS 2017, Savannah /USA, 22.-26.10.2017)]

c3- For pressure driven proportional release of reagents from blisters (Ref. 17) a piston-based approach is available (Microfluidic Chipshop, Blister Driver – ChipGenie® edition BD).

c4- For stick-packs in another pressure driven platform dosage has been accomplished by an elastomeric downstream valve (World Academy of Science, Engineering and Technology International Journal of Physical and Mathematical Sciences Vol:7, No:8, 2013, Long-Term On-Chip Storage and Release of Liquid Reagents for Diagnostic Lab-on-a-Chip Applications D. Czurratis, et al.).

c5- In addition, IQuum Inc (Allston, MA) patented the Liat Molecular Analyzer which is based on the lab-in-a-tube (Liat) technology. The Liat tube serves as reagent container. The flexible tube serves as contains all assay reagents pre-packed in tube segments, separated by peelable seal which are formed by a thermal weld of the plastic tube. By applying pressure to the tube segments adjacent to each seal, the seal can burst open to release reagents. In the Liat analyzer, multiple sample-processor modules are aligned with the Liat tube. Each module consists of an actuator and a CLAMP, whose positions can be controlled to manipulate a test sample within a tube. By synchronizing the motion of the actuators and clamps, various sample processes can be conducted within a tube. Such processes include ADJUSTING a LIQUIDS VOLUME in a segment; Stepwise RELEASING of a reagent to the adjacent segment; mixing; etc.

Responses:

Thank you for providing several useful references which have helped us to carefully revise the abstract and the introduction section. We have added 43 new lines to the old introduction. We realize that our introduction was poor in discussing different aspects of the state of art. Several liquid handling and liquid storage techniques have been separately developed that have accelerated the growth of microfluidic industry specially. Therefore, in our revised introduction we not only used the references given by the reviewer but also some other featured and state of art liquid handling and liquid storage. As a result, the revised manuscript thoroughly discusses the state

of art in the context of previous approaches to emphasize the novelty and advantages the micro-dispenser. We have also revised the abstract accordingly and in line with your other suggestions. The revised abstract reveals the advantages of the micro-dispenser in the light of previous state of art and states some of the key characteristics and applications of the micro-dispenser. The details of changes are as follows.

The changes in Abstract are as follows:

*a- **Line 18-25:** We have modified previously known line 20-22 and the revised statement can be found here. The revised abstract mentions some main state of art technologies in liquid handling and liquid storage technology. **Line 18-25 says:** “Yet, their success may depend on a generic low-cost device that provides on-chip storage and performs important fluidic handling tasks. To date, many different methods have been developed that separately cope with on-chip storage and fluidic handling e.g., wax, hydrophobic, capillary and siphon valves, pneumatic and micro-balloon pumping and stick packages. However, there is currently no microfluidic device that simultaneously achieves liquid storage and efficient, cost-effective and universal liquid dispensing on different microfluidics platforms.”*

*b- **Line 31-41:** Here, we have added some of the key micro-dispenser features such as shelf-life, characterization results and its applications. **Line 31-41 says:** “We demonstrate both long shelf-life and accurate, repeatable and controlled dispensing of reagents. The micro-dispenser shows excellent dispensing accuracy, especially on lab-on-a-disc platforms where the accuracy is comparable with conventional micro-dispensers. Shelf-life experiments performed demonstrate storage of di-water and ethanol for a simulated period of two-year and one-year with loss of 0.2% and less than 0.1%, respectively. As a proof of principle, a finger pressure actuating micro-dispenser is shown for low cost lab-on-chip systems and for separating plasma from blood cells in lab-on-Disc platforms. Further, we introduce a new method for processing bio assays. The micro-dispenser technology intends to accelerate the growth of the microfluidics industry and enable much-needed affordable point-of-care diagnostic services, specially at resource limited settings.”*

The changes in Introduction are as follows:

We have thoroughly discussed the advantages of our innovation in the light of different aspects of the state of the art. First, we thoroughly discuss the state of art in liquid handling techniques such as metering, aliquoting, valves and pumps and separating blood plasma. In the following parts we thoroughly discuss the state of art in liquid storage techniques and summarize our introduction with introducing our innovative technology and its advantages. The details of changes in abstract are as follows:

Regarding liquid handling:

*c1, c2- **Line 60-67:** Previously known line 34-44 has been completely modified. Further, we have extended this section and coherently discussed the state of art and significant findings in aliquoting, metering, valves and pumps in order to emphasize the novelty and advantages of our proposed method. In this section we not only used the relevant and useful references provided by the reviewer but also cited to other*

relevant state of art technologies. The new statements can be found at: **Line 60-67 says:** “Currently, a large variety of active and passive fluid handing techniques have been developed and used together in order to carrying out different fluidic tasks. Aliquoting and metering are generally performed in few consecutive steps by employing downstream micro structures and/or valves. For example, in LOD platforms reagents are first metered by dividing into several defined sub-volumes, and then transferred to different destinations ^{1,13,15}. Using additional micro-chambers and valves enables sequentially and time-dependently release of liquids in theses platforms ¹⁵.”

c3- Line 99-101: We have considered the release of reagents from blisters using a piston-based in our revised introduction. The new statement can be found at: **Line 99-101 says:** “A mechanical actuator can be used for more accurate and controlled release of the content stored in blisters, which of course adds complexity, costs and occupies space ²⁹.”

c4- Line 109-112: We have added a similar publication that demonstrate multiple releasing of reagents in stick packages. The new statement can be found at: **Line 109-112 says:** “The release efficiency of these packages has been improved by adding a receiving chamber and a membrane valve in the downstream that enables multiple releasing of reagents ^{30,31}.”
²⁹.”

c5- Line 81-85: We have discussed the development of lab-on-a-tube. **Line 81-85 says:** “The development of a lab-in-a-tube (Liat) system has demonstrated that a system with robust valves and pumps is capable of automating a large variety of clinical assays ²⁵. However, Liat consists of many mechanical parts and specifically made components such as springs, clamps and holders, that makes the approach less generic and may complicate its mass production.”

Comment 4:

Line 168: Fig. 3: The terms "Net volume" and "total volume" need to be defined. No error bars are provided for the net volume.

Response:

In order to implement your suggestion in line with the other comments we have conducted more experiments and presented the results in a table that consists the standard deviation, deviation from mean and coefficient of variation values. The new table can be found at:

Line 337-340: Here, Table 2 has been added and previously known figure 3 has been removed. **Table 2 is follows:**

Table 1 Micro-dispenser dispensing accuracy in LOD platforms; the experimental data for four sets of similar micro-dispensers, approximate dispensed volume corresponding to different rotational frequencies, SD: standard deviation, CV: coefficient of variation, σ is standard deviation of mean and S_r is repeatability standard deviation

Sample No.	285xg (nl)	306xg (nl)	329xg (nl)	352xg (nl)	376xg (nl)	400xg (nl)	426xg (nl)	S_r (nl)
1	70-75	70-75	70-75	80-85	80-85	70-75	85-90	1
2	70-75	70-75	70-75	70-75	70-75	70-75	70-75	0.2
3	70-75	70-75	70-75	70-75	70-75	70-75	70-75	0.2
4	70-75	70-75	70-75	70-75	70-75	70-75	70-75	0.2
SD_{max}	4	4	4	7.1	7.1	2.9	9.5	---
$CV_{max}\%$	2.4	2.4	2.4	11.5	9.4	4	12.4	---
$\sigma_{max}\%$	2.5	2.5	2.5	6.1	6.1	2.5	8.2	---

Comment 5:

Line 209, Fig. 4. The resolution of the image of the two-membrane micro-pipette is too poor, and details are hardly visible. In addition, it is not clear how the liquid is guided from the membrane outlet of the micro-pipette into the receiving channel. This should be depicted in more detail. Also, the micropipette integrated in the disk seems to be tilted, and the reason for this should be explained.

Response:

Line 453-458: The previously known figure 4 which is now Figure 6 is found here. We have changed the picture with a higher quality image of blood plasma separation and moved the unnecessary photos to the supplementary. **Figure 6 is as follows:**

Figure 1: Images of a rotating LOD with micro-dispenser insert enabling separation of blood components. a) micro-dispenser with two apertures covered with C-flex and latex membranes, b) schematic of the LOD platform used in the experiment c) before actuation of the micro-dispenser, d) dispensing blood plasma and blood cells to two different destination chambers, at different rotational frequencies due to the difference between the membrane properties of the micro-dispenser

Line 431-438: The explanation that clarifies the function of the micro-dispenser and its bent shape is found here. Briefly, we bent the micro-dispenser right after the plasma microchannel and made it hydrophobic in order to avoid the plasma flow through blood cells microchannel. **Line 431-438, reads as follows:**

“A tacky adhesive was used to lock the micro-dispenser to the Lab-disc platform as seen in the figure. The micro-dispenser is bent after the plasma microchannel and is made hydrophobic that avoids the plasma flow through blood cells microchannel. The micro-dispenser used in this experiment has two apertures covered with two membranes as seen in the inset of Figure 1 In order to show that the micro-dispenser can be positioned in various ways and to facilitate the guidance of dispensed liquids the micro-dispenser was bent when inserted on the disc.”

Regarding experimental data provided by the authors

Comment 6:

Although the analytical treatment is described in the article, no comparison between theoretical and experimental data (critical pressure) is made in the results / discussion section. This should be included.

Response:

Line 262: The revised version of Table 1 can be found here. We have added several more simulation cases in this table that can be now compared with our experimental data presented in newly added Figure 4. **The new Table 1 is as follows:**

Table 2: The critical pressure simulated for different flask and membrane radii

Inner radius R_i (mm)	Thickness t (mm)	Tube radius R (mm)	Critical pressure p_0 (kPa)	
			Case 1	Case 2
4.8	2.4	3.3	146	140
4.8	1.6	3.3	118	114
4.8	0.8	3.3	74	73
2.4	0.8	3.3	246	239
1.6	0.8	3.3	346	339
3.9	1.6	5.0	67	65
3.9	0.8	5.0	35	39

Line 376-381: Figure 4 is found here. **Figure 4 is as follows:**

Figure 2: Characterization of actuation pressure for a micro-dispenser (membrane made of latex), set i: a dispenser of outer diameter $R= 6.5$ mm is sheathed in membranes of fixed internal diameter and different thickness, ii: a dispenser of outer diameter $R= 6.5$ mm is sheathed in membranes of fixed thickness and different internal diameters, and iii: a dispenser of outer diameter $R= 10.1$ mm is sheathed in membranes of fixed internal diameter and different thickness.

Comment 7:

For the analytical description, the equation used to calculate the critical pressure \$p\$ should be provided as well (line 117).

Response:

We have modified the text so that it is now clear that the actual pressure from the included fluid is based on the integral of the centrifugal forces in the fluid ‘above’ the aperture, whereas the critical pressure which opens the membrane must be calculated from the precise geometry and material of the tube and membrane. We point to two methods for calculating the critical pressure, where one is mentioned as a qualitative plane-stain analysis (valid for an assumed infinitely long membrane) and one is based on more accurate simulations of the membrane in more detail; for the latter, some relevant results are given in Table 1.

Line 230-233: *The newly added statement is found here. Line 230-233 reads: “This can be calculated from an integral of the centrifugal forces from the fluid surface level z to the aperture level h , which based on the geometry in Figure 1 leads to a pressure from rotation as:*

$$p(z) = \frac{1}{2} \rho \omega^2 [(H - h)^2 - (H - z)^2]$$

Comment 8:

For a specified setup (volume, liquid, and membrane material) experimental characterization and statistical data evaluation should be performed more comprehensively, especially providing quantitative data (Tables). Typical qualification comprises the following: typical volume range, setting a specific target volume (nominal volume), resolution and calibration, determination of actual volume including precision, accuracy (systematic error), repeatability (random error) (inter- and intra-pipette run accuracy and CV, load-to-load accuracy and CV), influence of dead air volume (for operation by mechanical actuation), dead-liquid volume, amount of residual liquid), influence of the temperature and barometric pressure. These values should be compared to the requirement of laboratory devices (i.e. pipettes, dispensers, to which the abstract refers (see also ISO 8655 or alternative standards and quality assurance documents for qualification of pipettes).

Response:

Thank you very much for pointing out these vital issues. As a result of implementing these thoughtful and valuable comments our results section has improved remarkably. In fact, by implementing your comments we have greatly modified the entire results section, which we are sincerely grateful for that (several figures and tables have been added and the old ones improved).

We have carried out sets of experiments in order to characterize the micro-dispenser for specific configurations. For LOD platforms we have conducted large number of experiments in order to find out the dispensing range corresponding to sealing tightness. We show the dispensing characteristics of specific set of micro-dispensers by listing the experimental data, standard deviation, coefficient of variation and

variation of standard deviation of the mean values. Similarly, for LOC platforms we used a pressure controller and found the dispensing volume range for specific micro-dispensers. We present experimental data and the error values for specific micro-dispenser and compared them with conventional micro-pipette using ISO 8655. In addition, we have thoroughly discussed the differences between the performance of a micro-dispenser when used in a LOC and that when used in LOD platforms. Here, we explain the air amount in the micro-dispensers, the possibility of dead-volume, residuals and the amount of them. We have also discussed the effect of air amount, the dead volumes and amount of residual liquids both in LOC and LOD. The results are compared with those of conventional dispenser using ISO 8655 standards. The details of implementation of your comments are as follows:

Line 337-340: Table 2 is found here. This table points to the micro-dispenser controlled release performance in LOD platforms (see the table in our response to comment 4). It lists the experimental results, standard deviation, coefficient of variation and deviation from standard mean values for four sets of specific micro-dispensers inserted into a LOD platform.

Line 341-343: Table 3 that is found here. This table points to the dispensing accuracy for specific micro-dispensers that can be used in LOC platforms and actuated by finger pressure or piston-based mechanisms. It lists the experimental results and the errors for three sets of specific micro-dispensers that are actuated using a pressure controller. **Table 3 is as follows:**

Table 3: Micro-dispenser dispensing accuracy in pressure-driven systems; the experimental data for three sets of micro-dispensers, SD is standard deviation, σ is standard deviation of mean and Sr is repeatability standard deviation

Sample No.	Actuation pressure	n= 1 (μ l)	n= 2 (μ l)	n= 3 (μ l)	n= 4 (μ l)	n= 5 (μ l)	Mean (μ l)	SD	σ	Sr (μ l)	Ref. σ $\pm(\mu$ l) ³⁹	Ref. Sr $\pm(\mu$ l) ³⁹
1	80 (kPa)	2.0	2.1	1.8	1.7	1.8	1.9	0.2	0.15	0.1	0.08	0.04
2	90 (kPa)	9.3	9.0	10.3	10.2	10.2	9.8	0.6	0.54	0.2	0.12	0.08
3	100 (kPa)	12.4	11.2	11.3	11.2	11.2	11.5	0.5	0.47	0.1	0.13	0.08-0.1

Line 376-381: Figure 4 is found here. This figure demonstrates the experimental results and standard deviation error values of the characterization of actuation pressure for specific sets of micro-dispensers (see the figure in our response to comment 6). It illustrates the effects of sealing tightness, membrane thickness and the outer diameter of the container.

Line 271-287: Here, we have modified the previous description and have provided further details about the experimental procedure and the effective parameters on dispensing range of different micro-dispenser configurations. **Line 271-287 reads as follows:** "Here, we show the dispensing accuracy of a set of four micro-dispensers filled with 95% di-water i.e., ~ 5 % air volume. The micro-dispensers are made by piercing a hole on FEP sealed tubes of outer and inner diameter of 2 mm, 1 mm, respectively that contains 22 μ l di-water. The apertures on the tubes are covered by latex membranes of outer and inner diameter 2.4 mm, 0.8 mm, respectively. In order to measure the volume dispensed at each discharge, we measured the change in liquid volume inside the micro-dispensers after each discharge, see Figure s1 in

supplementary. Note that the amount of liquid dispensed at a given pressure for different micro-dispenser configurations ranges from nano to micro liters which depends on the amount of air inside the micro-dispenser, the elastic properties of the aperture covering membrane and the tightness of the sealing. The combination of membrane tightness and the low amount of air we used here allows for highly controllable and accurate liquid dispensing. In Table 1 we show the standard deviation and coefficient of variation errors at different spinning frequencies for a set of four micro-dispensers of the same configuration. The results show that the micro-dispenser has comparable accuracy as conventional micro-dispensers ³⁹ in LOD systems.”

Line 288-309: Here, you can find the brief description of experimental procedure for characterization of specific pressure driven micro-dispenser. Further, Table 3 values including the standard error values required of laboratory devices are explained. **Line 288-309 reads as follows:** “We investigated the dispensing accuracy on LOC systems by connecting the micro-dispensers to a pressure controller (Negano Keiki, model no. PC20) and experimentally obtained the actuation pressure, i.e., the pressure at which the first droplet of liquid is dispensed. Suppose that the aperture has a centerline, in all sets the membrane asymmetrically covers the aperture, which allows unidirectional dispensing. The pressure increases gradually and the dispensed liquid is measured after ~30 s. Knowing this pressure, we investigated dispensing accuracy of three sets of similar micro-dispensers under different pressures above the actuation pressure. Table 3 lists the experimental data for the dispensing accuracy of three micro-dispensers, made from 1 ml syringes and a latex membrane with internal diameter and thickness of 3.2 mm and 0.8 mm, respectively. The discrepancy between liquid volumes dispensed in different micro-dispensers is rather due the inequality in the length and the exact positioning of the membrane as they are cut and placed manually. The piston pipette standards published at the international organization for standardization (ISO 8655) defines that maximum acceptable measurement uncertainties for dispensing 1-10 μ l, 10-100 μ l and 100-1000 μ l are 0.05-0.12 μ l, 0.12-0.8 μ l and 0.8-8 μ l, respectively ³⁹. According to the same, acceptable coefficient of variation for the abovementioned dispensing ranges are 5-0.8, 0.8-0.4 and 0.4-0.3. Knowing that these errors are twice as large for a multichannel pipettes, values listed in Table 1 and Table 3 show that the micro-dispenser has a great potential to operate within the range of acceptable measurement errors, especially for the use for LOD platforms.”

Line 310-336: Here, the performance of micro-dispenser in LOD and LOC platforms and their effective parameters are thoroughly discussed. The parameters that are considered here, are the effect of air amount in both systems. The effect of container material in LOC platforms and the effect of liquid volume (liquid plug length) in LOD platforms are thoroughly discussed in order to provide a basic understanding for design and development of a micro-dispenser. **Line 310-336 reads as follows:** “In LOD platforms, two additional parameters contribute in a more sensitive actuation and precise liquid dispensing. These are the initial amount of air inside the micro-dispenser and the liquid plug length, which both counter the effect of centrifugal pressure. The initial amount of air gradually expands after each dispensing which is due to dispenser’s flask is made of rigid materials. The expansion of air develops a

negative pressure (partial vacuum), that will affect the dispensing mechanism. Also, in these platforms at the constant rotational frequency, the centrifugal pressure applied decreases as the liquid plug length decreases⁴⁰. This allows for dispensing only a given amount of liquid at a given rotational frequency and to dispense the same volume of liquid again we need to increase the rotational frequency. The magnitude of this increase in centrifugal pressure relates to the amount of gas inside the micro-dispenser, and the length of liquid plug. In general, the larger gas volume, the lower the negative pressure developed, and the shorter liquid plug, the lower centrifugal pressure applied. However, when the micro-dispenser's flask is rigid and the initial amount of air is quite low, a dead liquid volume may be considered that can be calculated when the standard air volume and liquid volume are known. In LOC platforms i.e., when actuated using finger pressure, no negative pressure is developed as the flask of the micro-dispenser is made or partly made of a flexible material that shrinks in response to the air pressure decrease. Therefore, the volume of liquid that can be dispensed directly correlates with the amount of air. Hence, as a rule of thumb, the same amount of air must be available to dispense the entire liquid using finger pressure. In this case, the accuracy of dispensing can be improved by indirectly pressurizing the micro-dispense using a screw-based system that provides more control in applying pressure. Also, modifying the micro-dispenser design e.g., using the same mechanisms used for volumetric dispensing pipette^{41,42}, can be an alternative approach. The micro-dispenser can also be actuated using a piston-based mechanism, which can provide more accurate dispensing.”

Line 345-375: Here, we briefly explain the experimental procedure and discuss the characterization data in order to provide a clear understanding of the micro-dispenser performance that can be used to reproduce, design and develop specific micro-dispensers. **Line 345-375 reads as follows:** “Parameters of the micro-dispensers that can be used to tweak/determine the actuation pressure are: the difference between the internal diameter of the elastic membrane and external diameter of the container ($R-R_i$), thickness of membrane (t), and elasticity of the membrane. In order to investigate the effects of t , ($R-R_i$) and R , we conducted three sets of triple experiments. In total, we fabricated 27 micro dispensers of different configurations for studying the effect of ($R-R_i$) and t and R . We fabricated these micro-dispensers using 1 ml (\varnothing 6.6 mm) and 2.5 ml (\varnothing 10.1 mm) standard syringes (purchased from BD Plastipak™). We used a 1.2 mm drill bit to pierce the outlet aperture on the syringe barrel and a scalpel to cut out finger flanges of [on] the syringe barrel. Next, we inserted the piston of the syringe, pushed it inwards and sealed the barrel of the syringe. Thus, the dispensers contain \sim 0.6 ml and \sim 2.4 ml for those fabricated from 1 ml and 2.5 ml syringes, respectively. After filling each dispenser with \sim 90 % di-water we sheath them into elastic membranes previously cut to seal the aperture. All the membranes used were cut from latex tubing purchased from Kent Elastomer Products Inc. Note, that all experiments have been conducted in the same manner with air occupying \sim 10 % of the dispenser volume in order to study the influences of the presence of air on the performance of the micro-dispensers. The micro-dispensers were connected to pressure controllers using Luer-to-Luer connectors. The experiments were conducted by gradually increasing the input pressure and

monitoring the membrane behavior using a desktop magnifier. After injection of air sufficient to reach the critical opening pressure for the membrane, more air has to be continuously injected to keep the needed pressure for continuous flow. At the point where the pressure inside is higher than what the membrane can tolerate, the micro-dispenser will dispense. Figure 2 shows the results of three set of experiments with membranes of the same material illustrating the effects of $(R-R_i)$, (t) and (R) . These results can be compared with Table 2 that represents the simulation results for different cases. In general, in case of a thicker membrane or tighter sealing more pressure is required for actuating the micro-dispenser as verified by simulation results in Table 2. Also, the larger the internal diameter of the dispensers the lower the pressure for actuation if the fitting is similarly tight.”

Note, that we did not study the effect of barometric pressure because we did not have the possibly. However, we believe that the effects of barometric pressure and the operating temperature can be considered as a designing parameter which may be treated by selecting appropriate membrane properties and sealing tightness conditions. The operating temperatures of various membranes and containers are available in mechanical handbooks that can be use as principles of selectivity of micro-dispenser components. Nevertheless, we incubated specific micro-dispenser configuration made of Teflon and latex (for membrane) at 110 °C, 80 °C for two hours and successfully tested the viability of the product.

Comment 9:

For the long-time storage tests, data of real-time tests should be added as far as available (those used for calibration, e.g. for the first six month). Also, more volatile liquids, such as ethanol should be tested and data provided.

Response:

Thanks again for your valuable comments. We have now modified this part of our results section according to your comments. We have added the calibration data that we used both for water and newly added ethanol 70 % and clearly explained how we obtained the coefficients required for calculating the simulated shelf-life. Here, we added the experimental data for incubating micro-dispensers filled with water for duration of 180 days at room temperature. We also added the experimental data of incubating ethanol 70 % for duration 35 days at room temperature and used Arrhenius equation to calculate the simulated shelf life. The details of changes in the manuscript is as follows:

Line 383-392: A fundamental guideline for cost-effective, application specific micro-dispenser components i.e., the container and the membrane. **Line 383-392 reads as follows:** “The permeation of gases through the micro-dispenser correlates with the penetrant and selectivity of materials of micro-dispenser components. In essence, the micro-dispenser can be manufactured with any biologically inert materials that offer low to excellent impermeability such as aluminum and glass and polymers that are widely used in food packaging industry^{43,44}. The permeability data and material properties available at membrane and glove industry can be used for choosing an appropriate membrane⁴⁵⁻⁵⁰. In addition, the micro-dispenser can be coated in order to reduce the penetration rate through the micro-dispenser⁴⁴. The different material

preference allows to manufacture application specific, temperature specific and/or country specific micro-dispensers in order to provide a customizable cost-effective product.”

Line 397-414: Details of experimental and simulated shelf-life for micro-dispensers filled with water and ethanol 70 %. **Line 397-414 reads as follows:** “We incubated micro-dispensers filled with di-water at 65 °C for 14 days. For calculation of the equivalent real time we used Arrhenius equation and the activation energy for permeation reported in the literature ⁵¹⁻⁵³. We also incubated samples similar to those kept in the oven at room temperature for 180 days and weighed them on a daily basis. The results show weight loss <0.1 % which conforms with literature data we used in the Arrhenius equation. Figure 3 shows that the average weight loss of seven micro-dispenser samples maintained at 14 days at 65 °C i.e., equivalent to over three years at 23 °C is 0.37 %. We also simulated a shelf-life test for ethanol 70 % as an example of a volatile substance. For calculation of the equivalent real time we used Arrhenius equation and experimentally measured the accelerated aging rate (i.e., temperature coefficient) to calculate accelerated aging time duration. For calculation of temperature coefficient, we maintained three different samples at 30 °C, 40 °C, 50 °C and 60 °C and compared their corresponding permeation rates. The simulated results show less than 0.1 % weight loss for the simulated period which equals to more than one-year at 27 °C. We evaluated our simulated results by maintaining 3 micro-dispenser samples at room temperature for a period of a 35 days and the results confirmed our simulated shelf-life test.”

Line 415-419: Figure 5 is found here which is a revised version of previously known Figure 3b. Error bars have been provided in the new figure. **Figure 5 is as follows:**

Figure 3: The shelf-life of micro-dispenser; the shelf-life of 7 different micro-dispensers containing an average volume of 475µl, where the flask is made of glass and the membrane of neoprene, the error bars show the standard deviation of the mean and coefficient of variation values

Comment 10:

For the blood plasma separation, the hematocrit of the sample, the separated blood volume, the separation time, the plasma yield and purity need to be provided as well, and the values should be compared to clinical standards or reference methods.

Response:

Here, we would like to bring the attention of the reviewer to our newly added section where we introduce a novel method for blood plasma separation using a micro-dispenser. In order to evaluate the blood plasma collected using micro-dispenser we used a hemocytometer and a spectrometer and measured the absorbance of red blood cells using 575 nm wavelength. A comparison of absorbance rate between the plasma separated by manually aspiration and the plasma collected using micro-dispenser shows that the later has 10 % less absorbance value. Our hemocytometer test shows no cell for set of three samples.

Line 479-514: A newly added section and the details of blood plasma quality tests are found here. **Line 479-514 reads as follows:**

“Interlocking dispensing action (IDA)

In general, an IDA can consist of several micro-dispensers that are inserted and locked into each other. Different reagents can be pre-stored inside of each dispenser and the internal surfaces of the dispensers can be specifically modified. In the simplest configuration an IDA consists of a micro-dispenser which is inserted in a larger receiving vessel. This configuration can be used for automating blood components separation using conventional centrifuge machines. The conventional blood plasma method in many laboratories is carried out by skilled operators that carefully aspirate the supernatant fluid i.e., plasma or serum after centrifuging whole blood according to standard protocols. In Figure 4, we show an IDA that consists of a dispenser that is inserted into a standard laboratory safe-lock collection tube. The micro-dispenser is also fabricated from standard laboratory safe-lock Eppendorf tubes. The micro-dispenser is filled with blood, the IDA is placed into a centrifuge machine and centrifuged according to the standard blood plasma separation protocols. The amount of plasma that can be collected is determined by the position of aperture on the inserted micro-dispenser. For example, assuming 50 % of whole blood consisted of plasma, if the whole position is at 40 % of the tube height only 80 % of plasma is collectable. Note that, as we explained before, the magnitude of centrifugal force applied to the blood volume is related to the blood plug length in tube, which allows dispensing a given volume of blood at a given relative centrifugal force (rcf). Figure 4 shows the extraction of 45 % of blood plasma from 700 μ l whole blood using the rcf required to dispense the entire plasma located above the micro-dispenser aperture. In order to investigate the purity of the plasma collected we used a hemocytometer chip and a spectrometer machine to measure the absorbance of red blood cells. We compared the spectrometer and hemocytometer results for the plasma collected using IDA with the plasma collected using the conventional centrifugation method. For three samples the hemocytometer results did not show any cells in the plasma collected using IDA, see Figure s2 in supplementary. The absorbance of red blood cells in the plasma sample using 575 nm wavelength shows that the blood plasma collected by IDA has 10 % lower deviation from the blank

control. Note that for the conventional centrifugation method we used Eppendorf Centrifuge 5810 R machine and spun the sample at 900xg for 10 minutes. We used the same setting for blood plasma separation using IDA but we increased the speed to 1400xg for approximately 2 minutes to stretch the membrane and dispense the plasma into a separate tube. Using the IDA concept, the entire plasma separation process is automatized and can also be integrated with further downstream analysis.”

Line 515-520: Newly added Figure 8 is found here. This figure introduces a novel method for automating clinical processes here exemplified by blood plasma separation. **Figure 8 is as follows:**

Figure 4: A configuration of interlocking dispensing analyzer (IDA) assembled to automated blood plasma separation in laboratories. IDA for blood plasma separation is constituted of a micro-dispenser made of a safe-lock Eppendorf tube that is interlocked into a larger safe-lock collection tube. a) an IDA before inserting in centrifuge machine, b) after centrifugation according to the standard protocols for 20 minutes c) blood cells in the micro-dispenser and blood plasma in a safe-lock Eppendorf tube.

Comments 11, 12:

-A manufacturing guideline should be provided, e.g. materials of the glass tubes, filling of the glass tubes, sealing procedure, implementation of aperture.

-The potential for large scale manufacturing should be critically discussed (part numbers, costs) and be compared to other established approaches of reagent prestorage and release.

Response:

Thank you for pointing this important issue. We have added a new section that thoroughly discusses the manufacturing method, the scalability, manufacturing cost and the filing of the proposed micro-dispenser and the potential for large scale manufacturing.

Line 150-189: Here, we provide a manufacturing guideline, and thoroughly explain the manufacturing method for research centers and a possible mass-production process, the scalability, manufacturing cost and the filing of the proposed micro-dispenser. **Line 150-189 reads as follows:** “The simplicity of the proposed micro-dispenser enables various methods of fabrication both for prototyping and in mass production. Figure 5a-d shows a method for fabricating a cylindrical micro-dispenser that is more suitable for producing prototypes at research centers and universities. The figure shows the components required for fabricating the device i.e., an

impermeable tube and a flexible membrane with low permeability and preferably high elasticity. The tube is measured, cut and a drill is used to pierce one or multiple apertures. Both sides of the tube are sealed using a clamp and a hot plate. The tube is sheathed in the membrane, which was previously measured and cut by a sharp scalpel, using a three-pronged plier. At this point the membrane is half-covering the aperture and the micro-dispenser is autoclaved and packed, if necessary. Before use, the reagent is dispensed into the tube and the membrane is pulled to now completely cover the aperture. Note that, the micro-dispenser can be filled using a syringe or an Eppendorf micro loader depending on the micro-dispenser dimensions. All components of the micro-dispenser are purchased from well-known suppliers that allows for estimating the fabrication cost of prototype micro-dispensers. Depending on the materials used the micro-dispenser costs approximately 2-6 \$ i.e., considering the average labor wage of European Union. For other configurations of micro-dispensers e.g., when we use safe-lock Eppendorf tubes or pipettes, prototyping costs will be slightly higher or lower depending of the material used for fabrication. For mass production, the micro-dispenser flask can be produced in bulk using different standard mass production methods e.g., injection molding for polymeric materials, molding for metals such as aluminum. The membrane raw material can be purchased from mass product suppliers and cut into precisely measured pieces using a robotic knife. The robotic knife can be connected to holders and conveyors that hold, transfer and measure the membrane for cutting. The micro-dispenser can be assembled by developing customized robotic arms that enable exact positioning of the membrane.

In Figure 5e we show how conventional tubing, pipette, pipette tips and storage tubes can all be used to manufacture micro-dispensers in various sizes. Note that micro-dispensers can be manufactured in different shapes as well. However, we only show cylindrical shapes because of the abundant availability of cylindrical vessels. The smaller micro-dispensers shown in subfigures, can be used directly or with slight modification as inserts in LOC and LOD platforms. The bigger micro-dispenser made from a pipette can be used in several different settings when an airtight dispenser that prevents air exposure to the liquid inside is needed. For example, this is highly desirable in medical dispensers such as eye and ear droppers. Based on the micro-dispenser concept, we have devised a new method for processing liquid handling in clinical analyses, see Figure 5e. The method is based on interlocking dispensing action (IDA), which we will explain its function for automating laborious clinical analyses in the next sections.”

Line 190-194: *In Figure 2a-d we illustrate a step-by-step fabrication method and in Figure 2e we demonstrate the scalability of the micro-dispenser by giving examples of micro-dispensers of different sizes. In Figure 2e, we show the scalability of the proposed micro-dispenser by showing different sizes of micro-dispenser. **Figure 2 is as follows:***

Figure 5: Scalability and procedure of producing a cylindrical micro-dispenser. a) drilling the outlet aperture on a predefined length of an impermeable tube; b) sealing each side of the tube, c) sheathe in a predefined length of an elastic membrane, d) close-up view of a micro-dispenser, e) examples of scalability of micro-dispensers

Further suggestions / discussion

Comment 13:

Suggestion: Line 1, The suggested micro-pipette could also be designated as “micro-dispenser” as it can only dispense, and aspirate liquids.

Response:

We appreciate the suggestion and have changed the micro-pipette to micro-dispenser throughout the manuscript.

Comment 14:

Suggestion: Line 26, Consider: “the approach is INTENDED to [...]”

Response:

Line 39-41: We agree and the modified statement can be found here. **Line 39-41 says:**

“The micro-dispenser technology intends to accelerate the growth of the microfluidics industry and enable much-needed affordable point-of-care diagnostic services, specially at resource limited settings.”

Comment 15:

Suggestion: Line 147, For details about the experimental procedure the electronic supplement should be referenced here.

Response:

We have added a section that provides a guideline for manufacturing, see our response to comments 11, 12. We have also revised our entire manuscript and referred to supplementary wherever required throughout the manuscript.

Comment 16:

Suggestion: Line 164, A comparison of the shelf-life between two systems appears only meaningful if two specific systems are compared. Both systems, the micro-pipettes, and the reference system could be probably produced from various materials.

Response: *We completely agree and have removed the comparison.*

Comment 17:

Suggestion: All error bars should be defined in the corresponding figure legends.

Response:

All error bars have been defined in the corresponding figure legends.

Reviewer#3

Comments-Overview:

It is certainly true that the two drawbacks of POC devices for field use addressed in this manuscript are relevant, and among the reasons why these devices are not commonly used to date. In this respect, the work presented is very relevant. This work builds well on previous work looking at other means to achieve liquid control and storage (e.g. blister packages) and is novel in its approach. The work would be more valuable if additional result data can be added. It is assumed that this data is not included in the confidential data mentioned. Specific points are outlined below...

General response to the reviewer:

We are grateful for your valuable comments. We have realized that the manuscript has had several shortcomings that were pointed out by you and the other reviewers. We have implemented your suggestions and as a result, our manuscript has become far better and much improved. In general, we have extended our introduction to nearly one page more in order to provide a comprehensive overview of the state of art in liquid dispensing and liquid storage in LOC devices. We have also added several new sections in our method, results and application sections. Specifically, in order to implement your comments about the use of micro-dispenser in resource limited settings, we have demonstrated the integration of a micro-dispenser with a specimen LOC device used finger pressure for actuation of the device. Further, we characterized specific pressure-driven micro-dispensers that can be used as a guideline for integrating micro-dispensers with LOC devices. We have also provided further information about the viability of shelf-life results and performed more experiments to show the shelf-life for volatile materials. We have also added all the dimensions of the micro-dispensers used throughout the manuscript. Furthermore, we have added two new Tables and four new figure and revised our previously presented tables and figures by adding more experimental data and adding error bars. Our detailed responds to your comments are as follows:

Comment 1:

Of particular interest would be the finger activation, as this would be very useful in resource limited settings. Presumably glass cannot be used for this. The authors should provide data showing repeatability for finger actuation of polymer-based devices. As one of the central themes of the work is repeatability, this is very important to show. How easy is finger actuation?

Response:

We have now added a new section that describes the integration of a micro-dispenser that can be actuated with a finger pressure. In this section we have used a micro-dispenser made of thin FEP tube that is connected to a LOC specimen.

Line 460-474: Here, we have provided a brief explanation of integrating a micro-dispenser with LOC devices. **Line 460-475 reads as follows:** "In resource limited settings, POC devices should be self-contained, run at low power, function in extreme meteorological conditions (Extreme Point of Care or EPOC) and ideally meet the ASSURED criteria ^{35,54-56}. An attractive POC test device that conforms with

ASSURED criteria must integrate micro-dispensing and liquid storage. The attempts at developing an integrated micro-dispenser-liquid storage for LOC devices started more than a decade ago, i.e., details can be found in different review articles ^{57,58}. But a single generic low-cost device that is able to solve problem of liquid storage and dispensing have never been presented. The micro-dispenser introduced here enables the integration of liquid storage and dispensing at a low-cost and is compatible with various techniques. As a proof of principle, here we integrate a micro-dispenser made from a flexible impermeable material with a LOC specimen made of poly (methyl methacrylate), see Figure 6. The figure shows a micro-dispenser made of a flexible tubing which is half-filled with black died di-water. A given amount of liquids is pumped to the micro-structure each time the micro-dispenser is actuated i.e., using finger pressure.”

Line 475-478: Here, Figure 7 is found. The figure illustrates the integration of a micro-dispenser with a LOC specimen and its actuation using finger pressure has been added. **Figure 2 is as follows:**

Figure 6 Micro-dispenser for integrating with LOC devices. a) a micro-dispenser connected to a micro-structure, b) actuating micro-dispenser by applying finger pressure c) applying further pressure d) stop applying pressure

Note, that we have also experimentally investigated the actuation pressure for specific configurations of micro-dispenser that can be used for integrating a micro-dispenser that can be actuated with finger pressure or a piston-based mechanism. We have verified our experimental data with our simulation results. **In these regards we would like to bring your attention to our response to reviewer#2's comments number 8, 11 and 12.**

Comment 2:

Authors provide flask inner radius, but not outer, and it is difficult to gauge how easy it is to use finger actuation. Very small diameter tubes can be very stiff.

Response:

*We have now added both the inner and the outer diameter of the micro-dispensers we used throughout the manuscript. Also, we would like to bring the reviewer's attention to our response to his/her **comment 1** where in Figure 7 we show the finger actuation.*

Comment 3:

The authors also mention low cost and scalability as advantages but provide no data on this. It would be useful to know what typical cost points are for this type of pipette, and also what is meant by scalability. Is it the ability to increase volume of device, or scaled manufacture of the device?

Response:

Thank you for pointing this important issue.

We have added a new section that thoroughly discusses the manufacturing method, the scalability, manufacturing costs and the filing of the proposed micro-dispenser and the potential for large scale manufacturing. The details of the newly added statements are as follows:

Line 150-189: *Here, we provide a manufacturing guideline, and thoroughly explain the manufacturing method for research centers and a possible mass-production process, the scalability, manufacturing cost and the filing of the proposed micro-dispenser. To read the new statements kindly refer to our response to the reviewer#2's comments 11, 12.*

Line 190-194: *In Figure 2a step-by-step we illustrate a fabrication method and in Figure 2b we demonstrate the scalability of the micro-dispenser by giving examples of micro-dispensers of different sizes. To see this figure kindly refer to our response to the reviewer#2's comments 11, 12.*

Comment 4:

Would it be possible to provide data in table 1 for the 1mm pipette referred to in fig 3, and also provide the volume of the 1mm pipette? It would appear that most volumes are several 10's to 100's of micro liters, and a graph showing aliquoting accuracy for the full volume of the pipette (e.g. 25ul or 475ul) is important.

Response:

We realize that this vital information had not been provided. We have now clearly described the details of the micro-dispensers including the volumes of the content liquid, the outer and the inner diameter of the containers and the membranes used in conducting various experiments and simulations. We have also added more simulation cases that verify our newly added experimental data presented in Figure 4. We have also removed Figure3a and added a Table 2 with more results and different error values.

Line 262: *We have modified Table 1 and added more results that can be verified with Figure 4. To see Table 1 kindly refer to our response to the comment 1.*

Line 337-340: Here, Table 2 has been added and previously known figure 3 has been removed. **To see Table 2 kindly refer to our response to the comment 4.**

Line 376-381: Figure 4 can be found here which characterization of actuation pressure for different sets micro-dispensers. The effect of membrane thickness, the effect of sealing tightness and the effect of the outer diameter of the container have been investigated. All the dimensions and error bars have been explained in the figure or the caption. **To see figure 4 kindly refer to our response to the comment 6.**

Comment 5:

It is also not clear how the viscosity of different fluids (expected in real use) would influence dispensing accuracy.

Response:

We have conducted a set of experiment to evaluate the effect of liquid properties on dispensing liquid. For this purpose, we used, di-water, methanol and a solution of 75 % glycerol and 25 % di-water. For specific micro-dispensers we used in our experiment our results indicate that while the viscosity has negligible effect the density of the material in centrifugal microfluidics has obvious effect in actuation of the micro-dispenser. However, we have not included this experiment in the current manuscript, we will add it if the reviewer urges on adding this.

Comment 6:

The authors need to motivate in more detail why their accelerated life time tests are valid in this device? Material used for the test (glass) is ideal for storage, but is not always favored in resource limited settings, so that data for polymer materials is required. Even though tests were done, this data is not presented. Additionally, tests are done with de-I water, and data for expected field reagents would be useful as these are often corrosive in nature.

Response:

Thank you for pointing these important issues. In order to calibrate our shelf-life simulation calculation we have incubated samples of the micro-dispensers filled with water at room temperature for 180 days. We have also added new experimental and simulated results for the shelf-life of a micro-dispenser filled with ethanol 70 %. Similarly, we incubated samples of the micro-dispensers filled with ethanol 70 % at room temperature for 35 days. We agree that glass is perhaps not the most suitable liquid storage for resource limited settings. However, we realized that the most vulnerable component of the micro-dispenser to permeation is the elastic membrane due to their relatively porous material structure. As such we confined our experiments to the use of glass as the container and studied the shelf-life of the micro-dispensers made of different membrane materials. Considering that the permeability rates of large variety of polymers to various gases are available in different standard, mechanical handbook like ASME. and in research articles about food storage and packaging. Now, we provide a fundamental guideline that helps researchers to select appropriate material in fabricating the micro-dispensers. The details of changes in manuscript are as follows:

Line 397-414: Details of experimental and simulated shelf-life for micro-dispensers filled with water and ethanol 70 %. **Line 397-414 reads as follows:** “We incubated

micro-dispensers filled with di-water at 65 °C for 14 days. For calculation of the equivalent real time we used Arrhenius equation and the activation energy for permeation reported in the literature⁵¹⁻⁵³. We also incubated samples similar to those kept in the oven at room temperature for 180 days and weighed them on a daily basis. The results show weight loss <0.1 % which conforms with literature data we used in the Arrhenius equation. Figure 3 shows that the average weight loss of seven micro-dispenser samples maintained at 14 days at 65 °C i.e., equivalent to over three years at 23 °C is 0.37 %. We also simulated a shelf-life test for ethanol 70 % as an example of a volatile substance. For calculation of the equivalent real time we used Arrhenius equation and experimentally measured the accelerated aging rate (i.e., temperature coefficient) to calculate accelerated aging time duration. For calculation of temperature coefficient, we maintained three different samples at 30 °C, 40 °C, 50 °C and 60 °C and compared their corresponding permeation rates. The simulated results show less than 0.1 % weight loss for the simulated period which equals to more than one-year at 27 °C. We evaluated our simulated results by maintaining 3 micro-dispenser samples at room temperature for a period of a 35 days and the results confirmed our simulated shelf-life test.”

Line 415-418: Previously known Figure 3b has been modified and the error bars have been provided. This figure is now presented as Figure 5. **To see figure 5 kindly, refer to our response to the comment 9.**

Comment 7:

Line 54: till should be until (till is seldom used)

Response:

Line 99-101: We have rephrased the sentence and removed till. **Line 99-101** reads as follows: “A mechanical actuator can be used for more accurate and controlled release of the content stored in blisters, which of course adds complexity, costs and occupies space²⁹.”

Comment 8:

Line 65: on-chip is not correct terminology for lateral flow. I would suggest on-test or leave out

Response:

We have removed this.

Comment 9:

Line 108: delta r should be delta R

Response:

We have corrected this and similar errors throughout the manuscript.

Comment 10:

Line 148: Nano = nano

Response:

We have corrected this.

Comment 11:

Line 153: significantly improved – can this be quantified

Response:

*We have removed this part; however, we have shown the effect of the sealing tightness for different micro-dispensers. The results are found at Figure 4, which can be found at **lines 376-381**. To see figure 4 kindly refer to our response to the comment 6.*

Comment 12:

Line 204,208: number of places have e.g. and then etc. I would suggest removing the etc. in all cases.

Response:

We have considered and removed these throughout the manuscript.

Comment 13:

References need reworking as are not all in same format.

Response:

We have considered and revised our references.

REVIEWERS' COMMENTS:

Reviewer #1 (Remarks to the Author):

The revised manuscript by Kazemzadeh et al. contains significant additional information describing and characterizing the operation and underlying principle of their microdispenser for storage and release of liquids. I appreciate these efforts and understand now better several aspects of the microdispenser. However, I still feel that the work does not represent a significant breakthrough in terms of storing and releasing liquids for bio-analytical applications. A key point from the authors is the idea of universality for their approach. Clearly, storage and dispensing of liquids has been demonstrated and used in point-of-care diagnostics using various approaches but why is universality so important, and is it truly achieved here? Digital PCR and droplet microfluidics are becoming commonly used nowadays; can the microdispenser in the manuscript handle non-miscible liquids and/or be used to generate droplets? Currently, blisters holding liquids or dried reagents are mostly used in devices and the limitations of existing technologies rather relate to sensitivity and specificity of tests. The coherence of the revised manuscript improved compared to the original version but the technical implementation of the microdispenser (e.g. figures 2, 6, 8) is insufficient to convince me that this concept can be scaled and manufactured in a technically compelling manner. The revised manuscript has technical merit but is more appropriate for publication in Scientific Reports rather than in Nature Communications.

Reviewer #2 (Remarks to the Author):

Dear editor, dear authors,

I acknowledge that the quality of the revised version of the manuscript "A micro-pipette for long-term storage and controlled release of liquids" has improved significantly.

The fabrication process is described in sufficient detail. The experimental data are much more comprehensive.

However, I still recommend the following minor revisions:

- 1) For historical reasons I strongly recommend to mention that the underlying principle of the described microdispenser is the well-known original Woods or Dunlop check valve US455899A, invented in 1891 (formerly used for bicycle tubes). For your information: a scheme is provided for instance in <https://de.wikipedia.org/wiki/Fahrradventil#Dunlopventil>). An image of the valve can

also be seen here: <https://www.sheldonbrown.com/images/woodsvalvecores.jpg>, which is embedded in the document <https://www.sheldonbrown.com/inner-tubes.html>

2) The hypothesis of the manuscript is (line 25 ff) “We report here on a UNIVERSAL micro-dispenser technique that incorporates long-term storage, valve and precise and effective aliquoting of reagents on microfluidics platforms. [...] As a proof of principle, a FINGER PRESSURE ACTUATING micro-dispenser is shown.” I recommend to better pronounce, that the precise aliquoting is not possible in all cases, e.g. when operated by finger pressure. Hence, with respect to “excellent dispensing accuracy” the claimed universality is not provided. I suggest to slightly adapt the corresponding wordings in the abstract, the introduction, and the conclusion.

3) Legend to new Fig. 5 (line 417). Please specify that the diagram shows the storage of WATER.

4) Line 83: “However, LIAT consists of many mechanical parts and specifically made components such as springs, clamps and holders, that makes the approach less generic and may complicate its mass production.” Please note: LIAT is being mass produced. The springs, camps etc. are not part of the disposable cartridge, they are rather part of the processing instrument.

5) Your response to Comment 5: “we have conducted a set of experiment to evaluate the effect of liquid properties ...”. I consider this piece of information extremely valuable. It should be included.

I recommend the manuscript for publication.

Reviewer #3 (Remarks to the Author):

The authors have put in considerable effort to address reviewer concerns. Particular concerns which I had have mostly been addressed satisfactorily.

Some concerns remain:

1. Lines 301-309 are not clear and it would be useful for the authors to give a better explanation here, especially how they are linking ISO standard to their own data. Although it is mentioned it is not explained and not immediately clear.

2. Table 2: the authors should explain more clearly how the data is obtained as it looks very similar and it is not clear why the experimental data is the same across all spin speeds and samples.

3. Would still be very useful to have data as supplemental information.

Response to referees

December 11th, 2018

REVIEWER REPORT and our response

Reviewer #1

Comments:

Reviewer #1 (Remarks to the Author):

The revised manuscript by Kazemzadeh et al. contains significant additional information describing and characterizing the operation and underlying principle of their microdispenser for storage and release of liquids. I appreciate these efforts and understand now better several aspects of the microdispenser. However, I still feel that the work does not represent a significant breakthrough in terms of storing and releasing liquids for bio-analytical applications. A key point from the authors is the idea of universality for their approach. Clearly, storage and dispensing of liquids has been demonstrated and used in point-of-care diagnostics using various approaches but why is universality so important, and is it truly achieved here? Digital PCR and droplet microfluidics are becoming commonly used nowadays; can the microdispenser in the manuscript handle non-miscible liquids and/or be used to generate droplets? Currently, blisters holding liquids or dried reagents are mostly used in devices and the limitations of existing technologies rather relate to sensitivity and specificity of tests. The coherence of the revised manuscript improved compared to the original version but the technical implementation of the microdispenser (e.g. figures 2, 6, 8) is insufficient to convince me that this concept can be scaled and manufactured in a technically compelling manner. The revised manuscript has technical merit but is more appropriate for publication in Scientific Reports rather than in Nature Communications.

General response:

Dear Reviewer, we sincerely thank you so much for reviewing our manuscript and for providing detailed suggestions and thoughtful comments. Kindly, find below our responses to your concerns and how we addressed your suggestions.

Comment:

A key point from the authors is the idea of universality for their approach. Clearly, storage and dispensing of liquids has been demonstrated and used in point-of-care diagnostics using various approaches but why is universality so important, and is it truly achieved here? Currently, blisters holding liquids or dried reagents are mostly used in devices and the limitations of existing technologies rather relate to sensitivity and specificity of tests.

Response:

We have removed the word universal in accordance with your concern. We would like to bring to the attention of the reviewer that although the numbers of different

solutions have been introduced to the problem of dispensing and storing, the commercially point-of-care devices using such dispensing and storage systems are either expensive or functionally limited. Furthermore, accurate dispensing and storage is currently not being combined into one system. In lab-on-a-chip based devices commercially available POC test are mostly lateral flow devices that are limited and need minor but necessary preparation step, e.g., see example, i-STAT by Abbott and VETSCAN FLEX4 Rapid Test by Abaxis. In lab-on-a-disc devices large incorporations have been using different techniques for storing and dispensing e.g., wax valves, and robotic pipettes for liquid handling and stick packages for storing liquids. This is due to the nature of these platform that requires strong normally-closed valves for preventing unexpected dispense of liquids due to centrifugation. In addition, the previously introduced techniques are not able to be opened and closed repeatedly and as a result aliquoting of liquids has been consuming large space of the lab-disc real-estate. The combination of these techniques reduces the number and variety of the diagnostic tests that can be carried out on a lab-disc cartridge and also increases the price of the entire system. In summary, our technology is unique in the following manners:

of liquids our approach is unique in different manners as explained below:

- a) it is a normally closed system that solves both storage and dispensing problems simultaneously; very similar to the function of a pipette. As such it allows developing automated sophisticated multi-steps diagnostic tests that usually need skilled operators to perform.
- b) It is simple and multifunctional and can combine different preparation steps such as separation, aliquoting and filtration without losing simplicity and imposing remarkable costs.
- c) It has low cost and is easy for mass fabrication and needs no special equipment for prototyping in research centers.

Comment:

Digital PCR and droplet microfluidics are becoming commonly used nowadays: can the microdispenser in the manuscript handle non-miscible liquids and/or be used to generate droplets?

Response:

Dear Reviewer, we do agree that digital PCR and droplet microfluidics are becoming attractive nowadays, but generating droplet for digital PCR and related applications is outside the scope of the current work. From application point of view, digital PCR is a last step in a laborious multi-step sample preparation where the cells/sample are isolated, lysed, and the nucleic acid material is extracted and fragmented before the emulsion is produced using droplet microfluidics. While in theory we would be able to generate droplets using our system, there are probably better ways to do this task and therefore it has not been the focus of our work. However, the ability to handle non-miscible is highly relevant question since our method should in theory work for any liquid with relevant viscosity. We have therefore performed experiments to test the storage and dispensing of oil and water and the results were satisfactory as expected. We thank the reviewer for bringing the important question about handling non-miscible fluids, and we believe our new experiment confirm this is the case.

Reviewer #2

Dear editor, dear authors,

I acknowledge that the quality of the revised version of the manuscript “A micro-pipette for long-term storage and controlled release of liquids” has improved significantly. The fabrication process is described in sufficient detail. The experimental data are much more comprehensive. 1) For historical reasons I strongly recommend to mention that the underlying principle of the described microdispenser is the well-known original Woods or Dunlop check valve US455899A, invented in 1891 (formerly used for bicycle tubes). For your information: a scheme is provided for instance in <https://de.wikipedia.org/wiki/Fahrradventil#Dunlopventil>. An image of the valve can also be seen here:

<https://www.sheldonbrown.com/images/woodsvalvecores.jpg>, which is embedded in the document <https://www.sheldonbrown.com/inner-tubes.html>

2) The hypothesis of the manuscript is (line 25 ff) “We report here on a UNIVERSAL micro-dispenser technique that incorporates long-term storage, valve and precise and effective aliquoting of reagents on microfluidics platforms. [...] As a proof of principle, a FINGER PRESSURE ACTUATING micro-dispenser is shown.” I recommend to better pronounce, that the precise aliquoting is not possible in all cases, e.g. when operated by finger pressure. Hence, with respect to “excellent dispensing accuracy” the claimed universality is not provided. I suggest to slightly adapt the corresponding wordings in the abstract, the introduction, and the conclusion.

3) Legend to new Fig. 5 (line 417). Please specify that the diagram shows the storage of WATER.

4) Line 83: “However, LIAT consists of many mechanical parts and specifically made components such as springs, clamps and holders, that makes the approach less generic and may complicate its mass production.” Please note: LIAT is being mass produced. The springs, camps etc. are not part of the disposable cartridge, they are rather part of the processing instrument.

5) Your response to Comment 5: “we have conducted a set of experiment to evaluate the effect of liquid properties ...”. I consider this piece of information extremely valuable. It should be included.

I recommend the manuscript for publication.

General Response:

Dear reviewer, we would like to sincerely thank you again for thoughtful and constructive comments and suggestions. Indeed, implementing your valuable comments has improved our manuscript significantly in all aspects. We have also modified our manuscript in order to address your remained concerns and comments. Kindly find below our point-by-point responses to your comments and suggestions.

Comment 1:

Recommendation:

For historical reasons I strongly recommend to mention that the underlying principle of the described microdispenser is the well-known original Woods or Dunlop check

[valve US455899A, invented in 1891 \(formerly used for bicycle tubes\). For your information: a scheme is provided for instance in <https://de.wikipedia.org/wiki/Fahrradventil#Dunlopventil>\). An image of the valve can also be seen here: <https://www.sheldonbrown.com/images/woodsvalvecores.jpg>, which is embedded in the document <https://www.sheldonbrown.com/inner-tubes.html>](https://de.wikipedia.org/wiki/Fahrradventil#Dunlopventil)

Response:

Dear reviewer, thank you for your comments. We believe that the micro-dispenser has similarities but is not exactly the same as the Woods check-valve. The micro-dispenser is sealed from both sides while the Woods check valve is open from either one or both ends. This difference creates a negative pressure inside the microdispenser which assists in generation of both airtight sealing and accurate liquid dispensing. The airtight seal mechanism is different in the Woods check valve, where the elastic membrane opens due to the increase in external applied pressure. Therefore, we believe that mentioning the Woods check valve for historical reasons is unnecessary for this publication.

Comment 2:

The hypothesis of the manuscript is (line 25 ff) “We report here on a UNIVERSAL micro-dispenser technique that incorporates long-term storage, valve and precise and effective aliquoting of reagents on microfluidics platforms. [...] As a proof of principle, a FINGER PRESSURE ACTUATING micro-dispenser is shown.” I recommend to better pronounce, that the precise aliquoting is not possible in all cases, e.g. when operated by finger pressure. Hence, with respect to “excellent dispensing accuracy” the claimed universality is not provided. I suggest to slightly adapt the corresponding wordings in the abstract, the introduction, and the conclusion.

Response:

We agree and have now removed the word universal and modified the manuscript accordingly. In abstract we have removed the word “universal” and edited our abstract. In introduction the word generic has been changed to integrated and we have modified our conclusion accordingly and addressed your comment.

Comment 3:

Legend to new Fig. 5 (line 417). Please specify that the diagram shows the storage of WATER.

Response:

Thank you for pointing this out. We have added di-water both to the figure caption and the figure title.

Comment 4:

Line 83: “However, LIAT consists of many mechanical parts and specifically made components such as springs, clamps and holders, that makes the approach less generic and may complicate its mass production.” Please note: LIAT is being mass produced. The springs, camps etc. are not part of the disposable cartridge, they are rather part of the processing instrument.

Response:

Thank you for enlightening the authors. We have modified the statement to address your comment and removed our comment on mass production.

5) Your response to Comment 5: “we have conducted a set of experiment to evaluate the effect of liquid properties ...”. I consider this piece of information extremely valuable. It should be included.
I recommend the manuscript for publication.

Response:

We have included this in the manuscript and it can be found at first paragraph, “Discussion” section, **lines 378-384**.

Reviewer#3

Remarks to the Author:

The authors have put in considerable effort to address reviewer concerns. Particular concerns which I had have mostly been addressed satisfactorily.

Some concerns remain:

1. Lines 301-309 are not clear and it would be useful for the authors to give a better explanation here, especially how they are linking ISO standard to their own data. Although it is mentioned it is not explained and not immediately clear.
2. Table 2: the authors should explain more clearly how the data is obtained as it looks very similar and it is not clear why the experimental data is the same across all spin speeds and samples.
3. Would still be very useful to have data as supplemental information.

General Response:

Dear reviewer, we would like to sincerely thank you again for your valuable and constructive comments. We have modified our manuscript in order to implement your remained concerns. We would like you to kindly find below our point-by-point responses to your comments.

Comment 1:

Lines 301-309 are not clear and it would be useful for the authors to give a better explanation here, especially how they are linking ISO standard to their own data. Although it is mentioned it is not explained and not immediately clear.

Response:

*Thank you for now we realize that more explanation and clarification is required here. We have clarified why we have compared our experimental results with the ISO 8655 standard data. We have added new statements in order to clarify why the comparison has been made. These changes can be found at second paragraph, “Results, Dispensing characteristics” section, **Lines 212-214 and 222-223**.*

Comment 2:

Table 2: the authors should explain more clearly how the data is obtained as it looks very similar and it is not clear why the experimental data is the same across all spin speeds and samples.

Response:

Thank you very much for pointing this issue. We have now added explanation on how we obtain the experimental data and also clarified why the results are very similar both at different spinning speeds and across the samples.

Obtaining data: We have now explained our method used for obtaining data in the main text. The newly added statement can be found at first paragraph at “Results, Dispensing characteristics” section, lines 184-191. For more information about this we would like to draw the reviewer’s attention to Supplementary Figure 1.

Similarity of the results at different spinning speeds: This has been indirectly discussed at previously lines 310-325, where the difference between the dispensing accuracy of lab-on-a-chip and lab-on-a-disc systems was discussed. Now we have furthered clarified why the results are very similar both at different spinning speeds and across the sample. Briefly, we believe that the main reason behind this can be the expansion of the air inside the micro-dispenser, which creates a counter pressure against the centrifugal pressure and allows for consistent release of liquids when increasing spinning speed. Also, in lab-on-a-disc systems, the positioning and the length of the membrane is less critical compared to lab-on-a-chip devices, which is due to centrifugal force. The previously lines 310-325 are now available at second paragraph, “Discussion” section lines 385-398 and the newly added statements can be found at first paragraph under “Results, Dispensing characteristics” lines:179-181 and 191-198.

Comment 3:

Would still be very useful to have data as supplemental information.

Response:

Thank you for your comment. We have added more details and clarification, in response to your previous comments where suited in the main text to assist the readers.